# Matrix Norm Estimation from a Few Entries

**Ashish Khetan**
Department of ISE
University of Illinois Urbana-Champaign
khetan2@illinois.edu

**Sewoong Oh**
Department of ISE
University of Illinois Urbana-Champaign
swoh@illinois.edu

## Abstract

Singular values of a data in a matrix form provide insights on the structure of the data, the effective dimensionality, and the choice of hyper-parameters on higher-level data analysis tools. However, in many practical applications such as collaborative filtering and network analysis, we only get a partial observation. Under such scenarios, we consider the fundamental problem of recovering various spectral properties of the underlying matrix from a sampling of its entries. We propose a framework of first estimating the Schatten $k$-norms of a matrix for several values of $k$, and using these as surrogates for estimating spectral properties of interest, such as the spectrum itself or the rank. This paper focuses on the technical challenges in accurately estimating the Schatten norms from a sampling of a matrix. We introduce a novel unbiased estimator based on counting small structures in a graph and provide guarantees that match its empirical performances. Our theoretical analysis shows that Schatten norms can be recovered accurately from strictly smaller number of samples compared to what is needed to recover the underlying low-rank matrix. Numerical experiments suggest that we significantly improve upon a competing approach of using matrix completion methods.

## 1   Introduction

Computing and analyzing the set of singular values of a data in a matrix form, which is called the spectrum, provide insights into the geometry and topology of the data. Such a spectral analysis is routinely a first step in general data analysis with the goal of checking if there exists a lower dimensional subspace explaining the important aspects of the data, which itself might be high dimensional. Concretely, it is a first step in dimensionality reduction methods such as principal component analysis or canonical correlation analysis.

However, spectral analysis becomes challenging in practical scenarios where the data is only partially observed. We commonly observe pairwise relations of randomly chosen pairs: each user only rates a few movies in recommendation systems, and each player/team only plays against a few opponents in sports. In other applications, we have more structured samples. For example, in a network analysis we might be interested in the spectrum of the adjacency matrix of a large network, but only get to see the connections within a small subset of nodes. Whatever the sampling pattern is, typical number of paired relations we observe is significantly smaller than the dimension of the data matrix. We study all such variations in sampling patterns for partially observed data matrices, and ask the following fundamental question: *can we estimate spectral properties of a data matrix from partial observations?* We build on the fact that several spectral properties of interest, such as the spectrum itself or the rank, can be estimated accurately via first estimating the *Schatten $k$-norms* of a matrix and then aggregating those norms to estimate the spectral properties. In this paper, we focus on the challenging task of estimating the Schatten $k$-norms defined as $\|M\|_k = (\sum_{i=1}^{d} \sigma_i(M)^k)^{1/k}$, where $\sigma_1(M) \geq \cdots \geq \sigma_d(M)$ are singular values of the data matrix $M \in \mathbb{R}^{d \times d}$. Once we obtain accurate estimates of Schatten $k$-norms, these estimates, as well as corresponding performance guarantees, can readily be translated into accurate estimates of the spectral properties of interest.

## 1.1 Setup

We want to estimate the Schatten $k$-norm of a positive semidefinite matrix $M \in \mathbb{R}^{d \times d}$ from a subset of its entries. The restriction to positive semidefinite matrices is for notational convenience, and our analyses, the estimator, and the efficient algorithms naturally generalize to any non-square matrices. Namely, we can extend our framework to bipartite graphs and estimate Schatten $k$-norm of any matrix for any even $k$. Let $\Omega$ denote the set of indices of samples we are given and let $\mathcal{P}_\Omega(M) = \{(i, j, M_{ij})\}_{(i,j) \in \Omega}$ denote the samples. With a slight abuse of notation, we used $\mathcal{P}_\Omega(M)$ to also denote the $d \times d$ sampled matrix:

$$\mathcal{P}_\Omega(M)_{ij} = \begin{cases} M_{ij} & \text{if } (i, j) \in \Omega \ , \\ 0 & \text{otherwise} \ , \end{cases}$$

and it should be clear from the context which one we refer to. Although we propose a framework that generally applies to any probabilistic sampling, it is necessary to propose specific sampling scenarios to provide tight analyses on the performance. Hence, we focus on *Erdös-Rényi sampling*.

There is an extensive line of research in low-rank matrix completion problems [3, 11], which addresses a fundamental question of how many samples are required to *complete* a matrix (i.e. estimate all the missing entries) from a small subset of sampled entries. It is typically assumed that each entry of the matrix is sampled independently with a probability $p \in (0, 1]$. We refer to this scenario as *Erdös-Rényi sampling*, as the resulting pattern of the samples encoded as a graph is distributed as an Erdös-Rényi random graph. The spectral properties of such an sampled matrix have been well studied in the literature [7, 1, 6, 11, 14]. In particular, it is known that the original matrix is close in spectral norm to the sampled one where the missing entries are filled in with zeros and properly rescaled under certain incoherence assumptions. This suggests using the singular values of $(d^2/|\Omega|)\mathcal{P}(M)$ directly for estimating the Schatten norms. However, in the sub-linear regime in which the number of samples $|\Omega| = d^2 p$ is comparable to or significantly smaller than the degrees of freedom in representing a symmetric rank-$r$ matrix, which is $dr - r^2$, the spectrum of the sampled matrix is significantly different from the spectrum of the original matrix as shown in Figure 1. We need to design novel estimators that are more sample efficient in the sub-linear regime where $d^2 p \ll dr$.

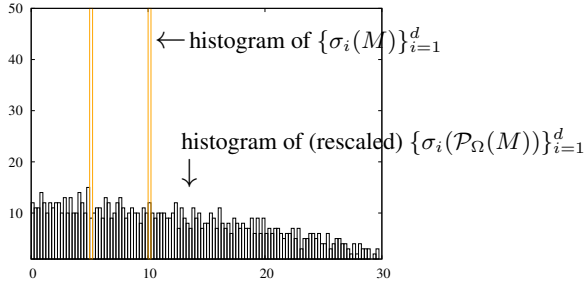

Figure 1: Histogram of (positive) singular values of $M$ with rank $r = 100$ (in yellow), and singular values of the sampled matrix (in black).

## 1.2 Summary of the approach and preview of results

We propose using an alternative expression of the Schatten $k$-norm for positive semidefinite matrices as the trace of the $k$-th power of $M$, i.e. $(\|M\|_k)^k = \text{Tr}(M^k)$. This sum of the entries along the diagonal of $M^k$ is the sum of total weights of all the closed walks of length $k$. Consider the entries of $M$ as weights on a complete graph $K_d$ over $d$ nodes (with self-loops). A closed walk of length $k$ is defined as a sequence of nodes $w = (w_1, w_2, \ldots, w_{k+1})$ with $w_1 = w_{k+1}$, where we allow repeated nodes and repeated edges. The *weight* of a closed walk $w = (w_1, \ldots, w_k, w_1)$ is defined as $\omega_M(w) \equiv \prod_{i=1}^k M_{w_i w_{i+1}}$, which is the product of the weights along the walk. It follows that

$$\|M\|_k^k = \sum_{w: \text{ all length } k \text{ closed walks}} \omega_M(w) \ . \tag{1}$$

Following the notations from enumeration of small simple cycles in a graph by [2], we partition this summation into those with the same pattern $H$ that we call a *$k$-cyclic pseudograph*. Let

$C_k = (V_k, E_k)$ denote the undirected simple cycle graph with $k$ nodes, e.g. $A_3$ in Figure 2 is $C_3$. We expand the standard notion of simple $k$-cyclic graphs to include multiedges and loops, hence the name *pseudograph*.

**Definition 1** We define an unlabelled and undirected pseudograph $H = (V_H, E_H)$ to be a $k$-*cyclic pseudograph* for $k \geq 3$ if there exists an onto node-mapping from $C_k = (V_k, E_k)$, i.e. $f : V_k \to V_H$, and a one-to-one edge-mapping $g : E_k \to E_H$ such that $g(e) = (f(u_e), f(v_e))$ for all $e = (u_e, v_e) \in E_k$. We use $\mathcal{H}_k$ to denote the set of all $k$-cyclic pseudographs. We use $c(H)$ to the number of different node mappings $f$ from $C_k$ to a $k$-cyclic pseudograph $H$.

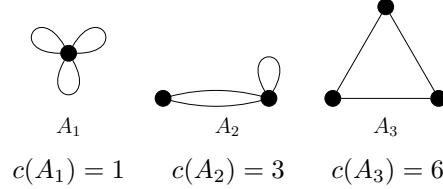

$$c(A_1) = 1 \qquad c(A_2) = 3 \qquad c(A_3) = 6$$

Figure 2: The 3-cyclic pseudographs $\mathcal{H}_3 = \{A_1, A_2, A_3\}$.

In the above example, each member of $\mathcal{H}_3$ is a distinct pattern that can be mapped from $C_3$. For $A_1$, it is clear that there is only one mapping from $C_3$ to $A_1$ (i.e. $c(A_1) = 1$). For $A_2$, one can map any of the three nodes to the left-node of $A_2$, hence $c(A_2) = 3$. For $A_3$, any of the three nodes can be mapped to the bottom-left-node of $A_3$ and also one can map the rest of the nodes clockwise or counter-clockwise, resulting in $c(A_3) = 6$. For $k \leq 7$, all the $k$-cyclic pseudo graphs are given in the Appendix E (See Figures 8–13).

Each closed walk $w$ of length $k$ is associated with one of the graphs in $\mathcal{H}_k$, as there is a unique $H$ that the walk is an Eulerian cycle of (under a one-to-one mapping of the nodes). We denote this graph by $H(w) \in \mathcal{H}_k$. Considering the weight of a walk $\omega_M(w)$, there are multiple distinct walks with the same weight. For example, a length-3 walk $w = (v_1, v_2, v_2, v_1)$ has $H(w) = A_2$ and there are 3 walks with the same weight $\omega(w) = (M_{v_1 v_2})^2 M_{v_2 v_2}$, i.e. $(v_1, v_2, v_2, v_1)$, $(v_2, v_2, v_1, v_2)$, and $(v_2, v_1, v_2, v_2)$. This multiplicity of the weight depends only on the structure $H(w)$ of a walk, and it is exactly $c(H(w))$ the number of mappings from $C_k$ to $H(w)$ in Definition 1. The total sum of the weights of closed walks of length $k$ can be partitioned into their respective pattern, which will make computation of such terms more efficient (see Section 2) and also de-biasing straight forward (see Equation (3)):

$$\|M\|_k^k = \sum_{H \in \mathcal{H}_k} \omega_M(H)\, c(H), \qquad (2)$$

where with a slight abuse of a notation, we let $\omega_M(H)$ for $H \in \mathcal{H}_k$ be the sum of all *distinct* weights of walks $w$ with $H(w) = H$, and $c(H)$ is the multiplicity of each distinct weight. This is an alternative tool for computing the Schatten norm without explicitly computing the $\sigma_i(M)$'s.

Given only the access to a subset of sampled entries, one might be tempted to apply the above formula to the sampled matrix with an appropriate scaling, i.e. $\|(d^2/|\Omega|)\mathcal{P}_\Omega(M)\|_k^k = (d^2/|\Omega|)\sum_{H \in \mathcal{H}_k} \omega_{\mathcal{P}_\Omega(M)}(H)\, c(H)$, to estimate $\|M\|_k^k$. However, this is significantly biased. To eliminate the bias, we propose rescaling each term in (1) by the inverse of the probability of sampling that particular walk $w$ (i.e. the probability that all edges in $w$ are sampled). A crucial observation is that, for any sampling model that is invariant under a relabelling of the nodes, this probability only depends on the pattern $H(w)$. In particular, this is true for Erdös-Rényi sampling. Based on this observation, we introduce a novel estimator that de-biases each group separately:

$$\widehat{\Theta}_k(\mathcal{P}_\Omega(M)) = \sum_{H \in \mathcal{H}_k} \frac{1}{p(H)} \omega_{\mathcal{P}_\Omega(M)}(H)\, c(H), \qquad (3)$$

where $p(H)$ is the probability the pattern $H$ is sampled. It immediately follows that this estimator is unbiased, i.e. $\mathbb{E}_\Omega[\widehat{\Theta}_k(\mathcal{P}_\Omega(M))] = \|M\|_k^k$, where the randomness is in $\Omega$. However, computing this estimate can be challenging. Naive enumeration over all closed walks of length $k$ takes time scaling as $O(d\,\Delta^{k-1})$, where $\Delta$ is the maximum degree of the graph. Except for extremely sparse graphs, this is impractical. Inspired by the work of [2] in counting short cycles in a graph, we introduce a novel and efficient method for computing the proposed estimate for small values of $k$.

**Proposition 2** *For a positive semidefinite matrix $M$ and any sampling pattern $\Omega$, the proposed estimate $\widehat{\Theta}_k(\mathcal{P}_\Omega(M))$ in (3) can be computed in time $O(d^\alpha)$ for $k \in \{3, 4, 5, 6, 7\}$, where $\alpha < 2.373$ is the exponent of matrix multiplication. For $k = 1$ or $2$, $\widehat{\Theta}_k(\mathcal{P}_\Omega(M))$ can be computed in time $O(d)$ and $O(d^2)$, respectively.*

This bound holds regardless of the degree, and the complexity can be even smaller for sparse graphs as matrix multiplications are more efficient. We give a constructive proof by introducing a novel algorithm achieving this complexity in Section 2. For $k \geq 8$, our approach can potentially be extended, but the complexity of the problem fundamentally changes as it is at least as hard as counting $K_4$ in a graph, for which the best known run time is $O(d^{\alpha+1})$ for general graphs [12].

We make the following contributions in this paper:

- We introduce in (3) a novel unbiased estimator of the Schatten $k$-norm of a positive semidefinite matrix $M$, from a random sampling of its entries. In general, the complexity of computing the estimate scales as $O(d\Delta^k)$ where $\Delta$ is the maximum degree (number of sampled entries in a column) in the sampled matrix. We introduce a novel efficient algorithm for computing the estimate in (3) exactly for small $k \leq 7$, which involves only matrix operations. This algorithm is significantly more efficient and has run-time scaling as $O(d^\alpha)$ independent of the degree and for all $k \leq 7$ (see Proposition 2) .

- Under the canonical Erdös-Rényi sampling, we show that the Schatten $k$-norm of an incoherent rank-$r$ matrix can be approximated within any constant multiplicative error, with number of samples scaling as $O(dr^{1-2/k})$ (see Theorem 1). In particular, this is strictly smaller than the number of samples necessary to complete the matrix, which scales as $O(dr \log d)$. Below this matrix completion threshold, numerical experiments confirm that the proposed estimator significantly outperforms simple heuristics of using singular values of the sampled matrices directly or applying state-of-the-art matrix completion methods (see Figure 4).

- Given estimation of first $K$ Schatten norms, it is straight forward to estimate spectral properties. We apply our Schatten norm estimates to the application of estimating the generalized rank studied in [20] and estimating the spectrum studied in [13]. We provide performance guarantees for both applications and provide experimental results suggesting we improve upon other competing methods. Due to space limitations, these results are included in Appendix B.

In the remainder, we provide an efficient implementation of the estimator (3) for small $k$ in Section 2. In Section 3, we provide a theoretical analysis of our estimator.

## 1.3  Related work

Several Schatten norm estimation problems under different resource constrained scenarios have been studied. However, those approaches assume specific noisy observations which allow them to use the relation $\mathbb{E}\left[\|f(M)g\|_2^2\right] = \sum_i f(\sigma_i(M))^2$ which holds for a standard i.i.d. Gaussian $g \sim \mathcal{N}(0, \mathbb{I})$ and any polynomial function $f(\cdot)$. This makes the estimation significantly easier than our setting, and none of those algorithms can be applied under our random sampling model. In particular, counting small structure for de-biasing is not required. [20, 8] and [9] propose multiplying Gaussian random vectors to the data matrix, in order to reduce communication and/or computation. [13] proposes an interesting estimator for the spectrum of the covariance matrix from samples of random a vector. [15] propose similar estimators for Schatten norms from random linear projections of a data matrix, and [16] study the problem for sparse data matrices in a streaming model.

One of our contribution is that we propose an efficient algorithm for computing the weighted counts of small structures in Section 2, which can significantly improve upon less sample-efficient counterpart in, for example, [13]. Under the setting of [13] (and also [15]), the main idea of the estimator is that the weight of each length-$k$ cycle in the observed empirical covariance matrix $(1/n)\sum_{i=1}^n X_i X_i^T$ provides an unbiased estimator of $\|\mathbb{E}[XX^T]\|_k^k$. One prefers to sum over the weights of as many cycles as computationally allowed in order to reduce the variance. As counting all cycles is in general computationally hard, they propose counting only increasing cycles (which only accounts for only 1/k! fraction of all the cycles), which can be computed in time $O(d^\alpha)$. If one has an efficient method to count all the (weighted) cycles, then the variance of the estimator could potentially decrease by an order of $k!$. For $k \leq 7$, our proposed algorithm in Section 2 provides exactly such an estimator.

We replace [13, Algorithm 1] with ours, and run the same experiment to showcase the improvement in Figure 3, for dimension $d = 2048$ and various values of number of samples $n$ comparing the multiplicative error in estimating $\|\mathbb{E}[XX^T]\|_k^k$, for $k = 7$. With the same run-time, significant gain is achieved by simply substituting our proposed algorithm for counting small structures, in the sub-routine. In general, the efficient algorithm we propose might be of independent interest to various applications, and can directly substitute (and significantly improve upon) other popular but less efficient counterparts.

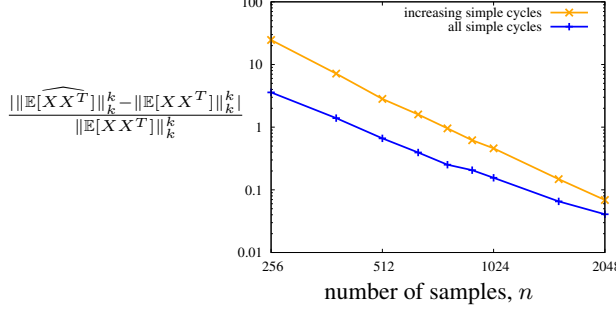

Figure 3: By replacing [13, Algorithm 1] that only counts increasing cycles with our proposed algorithm that counts all cycles, significant gain is acheived in estimating $\|\mathbb{E}[XX^T]\|_k^k$, for $k = 7$.

The main challenge under our sampling scenario is that existing counting methods like that of [13] cannot be applied, regardless of how much computational power we have. Under the matrix completion scenario, we need to $(a)$ sum over all small structures $H \in \mathcal{H}_k$ and not just $C_k$ as in [13]; and $(b)$ for each structure we need to sum over all subgraphs with the same structure and not just those walks whose labels form a monotonically increasing sequence as in [13].

## 2   Efficient Algorithm

In this section we give a constructive proof of Proposition 2. In computing the estimate in (3), $c(H)$ can be computed in time $O(k!)$ and suppose $p(H)$ has been computed (we will explain how to compute $p(H)$ for Erös-Rényi sampling in Section 3). The bottleneck then is computing the weights $\omega_{\mathcal{P}_{\Omega(M)}}(H)$ for each $H \in \mathcal{H}_k$. Let $\gamma_M(H) \equiv \omega_M(H)c(H)$. We give matrix multiplication based equations to compute $\gamma_M(H)$ for every $H \in \mathcal{H}_k$ for $k \in \{3, 4, 5, 6, 7\}$. This establishes that $\gamma_M(H)$, and hence $\omega_M(H)$, can be computed in time $O(d^\alpha)$, proving Proposition 2.

For any matrix $A \in \mathbb{R}^{d \times d}$, let $\mathrm{diag}(A)$ to be a diagonal matrix such that $(\mathrm{diag}(A))_{ii} = A_{ii}$, for all $i \in [d]$ and $(\mathrm{diag}(A))_{i,j} = 0$, for all $i \neq j \in [d]$. For a given matrix $M \in \mathbb{R}^{d \times d}$, define the following: $O_M$ to be matrix of off-diagonal entries of $M$ that is $O_M \equiv M - \mathrm{diag}(M)$ and we let $D_M \equiv \mathrm{diag}(M)$. Let $\mathrm{tr}(A)$ denote trace of $A$, that is $\mathrm{tr}(A) = \sum_{i \in [d]} A_{ii}$, and let $A * B$ denote the standard matrix multiplication of two matrices $A$ and $B$ to make it more explicit. Consider computing $\gamma_M(H)$ for $H \in \mathcal{H}_3$ as labeled in Figure 2:

$$\gamma_M(A_1) = \mathrm{tr}(D_M * D_M * D_M) \tag{4}$$
$$\gamma_M(A_2) = 3\,\mathrm{tr}(D_M * O_M * O_M) \tag{5}$$
$$\gamma_M(A_3) = \mathrm{tr}(O_M * O_M * O_M) \tag{6}$$

The first weighted sum $\gamma_M(A_1)$ is sum of all weights of walks of length 3 that consists of three self-loops. One can show that $\gamma_M(A_1) = \sum_{i \in [d]} M_{ii}^3$, which in our matrix operation notations is (4). Similarly, $\gamma_M(A_3)$ is the sum of weights of length 3 walks with no self-loop, which leads to (6). $\gamma_M(A_2)$ is the sum of weights of length 3 walks with a single self-loop, which leads to (5). The factor 3 accounts for the fact that the self loop could have been placed at various positions.

Similarly, for each $k$-cyclic pseudographs in $\mathcal{H}_k$ for $k \leq 7$, computing $\gamma_M(H)$ involves a few matrix operations with run-time $O(d^\alpha)$. We provide the complete set of explicit expressions in Appendix F. A MATLAB implementation of the estimator (3), that includes as its sub-routines the computation of the weights of all $k$-cyclic pseudographs, is available for download at

 The explicit formulae in Appendix F together with the implementation in the above url might be of interest to other problems involving counting small structures in graphs.

For $k = 1$, the estimator simplifies to $\widehat{\Theta}_k(\mathcal{P}_\Omega(M)) = (1/p) \sum_i \mathcal{P}_\Omega(M)_{ii}$, which can be computed in time $O(d)$. For $k = 2$, the estimator simplifies to $\widehat{\Theta}_k(\mathcal{P}_\Omega(M)) = (1/p) \sum_{i,j} \mathcal{P}_\Omega(M)_{ij}^2$, which can be computed in time $O(|\Omega|)$. However, for $k \geq 8$, there exists walks over $K_4$, a clique over 4 nodes, that cannot be decomposed into simple computations involving matrix operations. The best known algorithm for a simpler task of counting $K_4$ has run-time scaling as $O(d^{\alpha+1})$, which is fundamentally different.

---

**Algorithm 1** Schatten $k$-norm estimator

---

**Require:** $\mathcal{P}_\Omega(M), k, \mathcal{H}_k, p(H)$ for all $H \in \mathcal{H}_k$
**Ensure:** $\widehat{\Theta}_k(\mathcal{P}_\Omega(M))$
 1: **if** $k \leq 7$ **then**
 2:      For each $H \in \mathcal{H}_k$, compute $\gamma_{\mathcal{P}_\Omega(M)}(H)$ using the formula from Eq. (4)–(6) for $k = 3$ and Eq. (43) – (186) for $k \in \{4, 5, 6, 7\}$
 3:      $\widehat{\Theta}_k(\mathcal{P}_\Omega(M)) \leftarrow \sum_{H \in \mathcal{H}_k} \frac{1}{p(H)} \gamma_{\mathcal{P}_\Omega(M)}(H)$
 4: **else**
 5:      $\widehat{\Theta}_k(\mathcal{P}_\Omega(M)) \leftarrow$ Algorithm 2$[\mathcal{P}_\Omega(M), k, \mathcal{H}_k, p(H)$ for all $H \in \mathcal{H}_k]$         [Appendix A]
 6: **end if**

---

## 3 Performance guarantees

Under the stylized but canonical Erdös-Rényi sampling, notice that the probability $p(H)$ that we observe all edges in a walk with pattern $H$ is

$$p(H) = p^{m(H)}, \tag{7}$$

where $p$ is the probability an edge is sampled and $m(H)$ is the number of distinct edges in a $k$-cyclic pseudograph $H$. Plugging in this value of $p(H)$, which can be computed in time linear in $k$, into the estimator (3), we get an estimate customized for Erdös-Rényi sampling. Given a rank-$r$ matrix $M$, the difficulty of estimating properties of $M$ from sampled entries is captured by the *incoherence* of the original matrix $M$, which we denote by $\mu(M) \in \mathbb{R}$ [3]. Formally, let $M \equiv U\Sigma U^\top$ be the singular value decomposition of a positive definite matrix where $U$ is a $d \times r$ orthonormal matrix and $\Sigma \equiv \text{diag}(\sigma_1, \cdots, \sigma_r)$ with singular values $\sigma_1 \geq \sigma_2 \geq \cdots \geq \sigma_r > 0$. Let $U_{i,r}$ denote the $i$-th row and $j$-th column entry of matrix $U$. The incoherence $\mu(M)$ is defined as the smallest positive value $\mu$ such that the following holds:

     A1. For all $i \in [d]$, we have $\sum_{a=1}^r U_{ia}^2 (\sigma_a/\sigma_1) \leq \mu r/d$.

     A2. For all $i \neq j \in [d]$, we have $|\sum_{a=1}^r U_{ia} U_{ja} (\sigma_a/\sigma_1)| \leq \mu\sqrt{r}/d$.

The incoherence measures how well spread out the matrix is and is a common measure of difficulty in completing a matrix from random samples [3, 11].

### 3.1 Performance guarantee

For any $d \times d$ positive semidefinite matrix $M$ of rank $r$ with incoherence $\mu(M) = \mu$ and the effective condition number $\kappa = \sigma_{\max}(M)/\sigma_{\min}(M)$, we define

$$\rho^2 \equiv (\kappa\mu)^{2k} g(k) \max\left\{1, \frac{(dp)^{k-1}}{d}, \frac{r^k p^{k-1}}{d^{k-1}}\right\}, \tag{8}$$

such that the variance of our estimator is bounded by $\text{Var}(\widehat{\Theta}(\mathcal{P}_\Omega(M))/\|M\|_k^k) \leq \rho^2 (r^{1-2/k}/dp)^k$ as we show in the proof of Theorem 1 in Section D.1. Here, $g(k) = O(k!)$.

**Theorem 1 (Upper bound under the Erdös-Rényi sampling)** *For any integer $k \in [3, \infty)$, any $\delta > 0$, any rank-$r$ positive semidefinite matrix $M \in \mathbb{R}^{d \times d}$, and given i.i.d. samples of the entries of $M$ with probability $p$, the proposed estimate of* (3) *achieves normalized error $\delta$ with probability bounded by*

$$\mathbb{P}\left( \frac{\left| \widehat{\Theta}_k(\mathcal{P}_\Omega(M)) - \|M\|_k^k \right|}{\|M\|_k^k} \geq \delta \right) \leq \frac{\rho^2}{\delta^2}\left( \frac{r^{1-2/k}}{dp} \right)^k . \tag{9}$$

Consider a typical scenario where $\mu$, $\kappa$, and $k$ are finite with respect to $d$ and $r$. Then the Chebyshev's bound in (9) implies that the sample $d^2 p = O(dr^{1-2/k})$ is sufficient to recover $\|M\|_k^k$ up to arbitrarily small multiplicative error and arbitrarily small (but strictly positive) error probability. This is strictly less than the known minimax sample complexity for recovering the entire low-rank matrix, which scales is $\Theta(rd \log d)$. As we seek to estimate only a property of the matrix (i.e. the Schatten $k$-norm) and not the whole matrix itself, we can be more efficient on the sample complexity by a factor of $r^{2/k}$ in rank and a factor of $\log d$ in the dimension. We emphasize here that such a gain can only be established using the proposed estimator based on the structure of the $k$-cyclic pseudographs. We will show empirically that the standard matrix completion approaches fail in the critical regime of samples below the recovery threshold of $O(rd \log d)$.

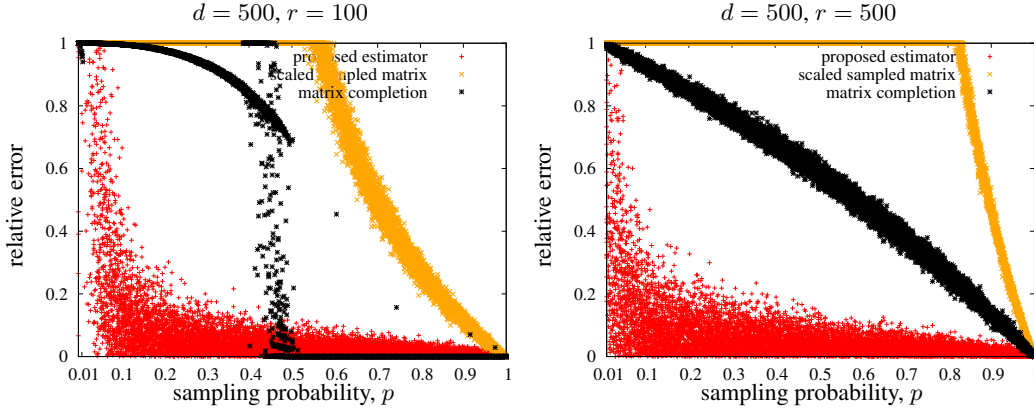

Figure 4: The proposed estimator outperforms both baseline approaches below the matrix completion threshold. For $k = 5$, comparison of the absolute relative error in estimated Schatten norm that is $\left| \|M\|_k^k - \widehat{\|M\|_k^k} \right| / \|M\|_k^k$ for the three algorithms: (1) the proposed estimator, $\widehat{\|M\|_k^k} = \widehat{\Theta}_k(\mathcal{P}_\Omega(M))$, (2) Schatten norm of the scaled sampled matrix, $\widehat{\|M\|_k^k} = \|(1/p)\mathcal{P}_r(\mathcal{P}_\Omega(M))\|_k^k$, (3) Schatten norm of the completed matrix, $\widetilde{M} = \text{AltMin}(\mathcal{P}_\Omega(M))$ from [10], $\widehat{\|M\|_k^k} = \|\widetilde{M}\|_k^k$, where $\mathcal{P}_r(\cdot)$ is the standard best rank-$r$ projection of a matrix. $\Omega$ is generated by Erdös-Rényi sampling of matrix $M$ with probability $p$.

Figure 4 is a scatter plot of the absolute relative error in estimated Schatten $k$-norm, $\left| \|M\|_k^k - \widehat{\|M\|_k^k} \right| / \|M\|_k^k$, for $k = 5$, for three approaches: the proposed estimator, Schatten norm of the scaled sampled matrix (after rank-$r$ projection), and Schatten norm of the completed matrix, using state-of-the-art alternating minimization algorithm [10]. All the three estimators are evaluated 20 times for each value of $p$. $M$ is a symmetric positive semi-definite matrix of size $d = 500$, and rank $r = 100$ (left panel) and $r = 500$ (right panel). Singular vectors $U$ of $M = U\Sigma U^\top$, are generated by QR decomposition of $\mathcal{N}(0, \mathbb{I}_{d \times d})$ and $\Sigma_{i,i}$ is uniformly distributed over $[1, 2]$. For a low rank matrix on the left, there is a clear critical value of $p \simeq 0.45$, above which matrix completion is exact with high probability. However, this algorithm knows the underlying rank and crucially exploits the fact that the underlying matrix is exactly low-rank. In comparison, our approach is agnostic to the low-rank assumption but finds the accurate estimate that is adaptive to the actual rank in a data-driven manner. Using the first $r$ singular values of the (rescaled) sampled matrix fails miserably for all regimes (we truncate the error at one for illustration purposes). In this paper, we are interested in the

regime where exact matrix completion is impossible as we do not have enough samples to exactly recover the underlying matrix: $p \leq 0.45$ in the left panel and all regimes in the right panel.

The sufficient condition of $d^2 p = O(dr^{1-2/k})$ in Theorem 1 holds for a broad range of parameters where the rank is sufficiently small $r = O(d^{k/((k-1)(k-2))})$ (to ensure that the first term in $\rho^2$ dominates). However, the following results in Figure 5 on numerical experiments suggest that our analysis holds more generally for all regimes of the rank $r$, even those close to $d$. $M$ is generated using settings similar to that of Figure 4. Empirical probabilities are computed by averaging over 100 instances.

One might hope to tighten the Chebyshev bound by exploiting the fact that the correlation among the summands in our estimator (3) is weak. This can be made precise using recent result from [18], where a Bernstein-type bound was proved for sum of polynomials of independent random variables that are weakly correlated. The first term in the bound (10) is the natural Bernstein-type bound corresponding to the Chebyshev's bound in (9). However, under the regime where $k$ is large or $p$ is large, the correlation among the summands become stronger, and the second and third term in the bound (10) starts to dominate. In the typical regime of interest where $\mu$, $\kappa$, $k$ are finite, $d^2 p = O(dr^{1-2/k})$, and sufficiently small rank $r = O(d^{k/((k-1)(k-2))})$, the error probability is dominated by the first term in the right-hand side of (10). Neither one of the two bounds in (9) and (10) dominates the other, and depending on the values of the problem parameters, we might want to apply the one that is tighter. We provide a proof in Section D.2.

**Theorem 2** *Under the hypotheses of Theorem 1, the error probability is upper bounded by*

$$\mathbb{P}\left( \frac{\left| \widehat{\Theta}_k(\mathcal{P}_\Omega(M)) - \|M\|_k^k \right|}{\|M\|_k^k} \geq \delta \right) \leq$$

$$e^2 \max \left\{ e^{-\frac{\delta^2}{\rho^2} \left( \frac{dp}{r^{1-2/k}} \right)^k}, e^{-(dp)\left( \frac{\delta d}{\rho r^{k-1}} \right)^{1/k}}, e^{-(dp)\left( \frac{\delta d}{\rho r^{k-1}} \right)}, e^{-\frac{\delta dp}{\rho}} \right\}. \qquad (10)$$

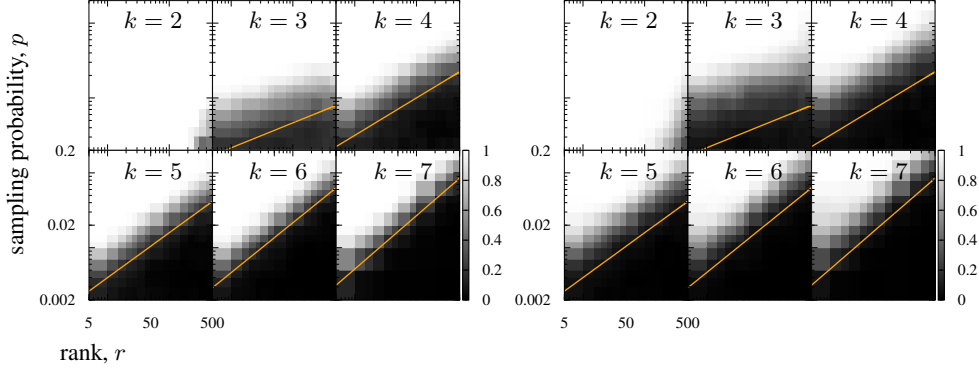

Figure 5: Each colormap in each block for $k \in \{2, 3, 4, 5, 6, 7\}$ show empirical probability of the event $\left\{ \left| \|M\|_k^k - \widehat{\Theta}_k(\mathcal{P}_\Omega(M)) \right| / \|M\|_k^k \leq \delta \right\}$, for $\delta = 0.5$ (left panel) and $\delta = 0.2$ (right panel). $\Omega$ is generated by Erdös-Rényi sampling of matrix $M$ with probability $p$ (vertical axis). $M$ is a symmetric positive semi-definite matrix of size $d = 1000$. The solid lines correspond to our theoretical prediction $p = (1/d)r^{1-2/k}$.

These two results show that the sample size of $d^2 p = O(dr^{1-2/k})$ is sufficient to estimate a Schatten $k$-norm accurately. In general, we do not expect to get a universal upper bound that is significantly tighter for all $r$, because for a special case of $r = d$, the following corollary of [15, Theorem 3.2] provides a lower bound; it is necessary to have sample size $d^2 p = \Omega(d^{2-4/k})$ when $r = d$. Hence, the gap is at most a factor of $r^{2/k}$ in the sample complexity.

**Corollary 1** *Consider any linear observation $X \in \mathbb{R}^n$ of a matrix $M \in \mathbb{R}^{d \times d}$ and any estimate $\theta(X)$ satisfying $(1 - \delta_k)\|M\|_k^k \leq \theta(X) \leq (1 + \delta_k)\|M\|_k^k$ for any $M$ with probability at least $3/4$, where $\delta_k = (1.2^k - 1)/(1.2^k + 1)$. Then, $n = \Omega(d^{2-4/k})$.*

For $k \in \{1, 2\}$, precise bounds can be obtained with simpler analyses. In particular, we have the following remarks, whose proof follows immediately by applying Chebyshev's inequality and Bernstien's inequality along with the incoherence assumptions.

**Remark 3** *For $k = 1$, the probability of error in (9) is upper bounded by* $\min\{\nu_1, \nu_2\}$, *where*

$$\nu_1 \equiv \frac{1}{\delta^2} \frac{(\kappa\mu)^2}{dp} \;, \quad and \quad \nu_2 \equiv 2\exp\Big( \frac{-\delta^2}{2}\Big( \frac{(\kappa\mu)^2}{dp} + \delta\frac{(\kappa\mu)}{3dp} \Big)^{-1} \Big) \,.$$

**Remark 4** *For $k = 2$, the probability of error in (9) is upper bounded by* $\min\{\nu_1, \nu_2\}$, *where*

$$\nu_1 \equiv \frac{1}{\delta^2} \frac{(\kappa\mu)^4}{d^2 p}\Big(2 + \frac{r^2}{d}\Big) \;, and \quad \nu_2 \equiv 2\exp\Big( -\frac{\delta^2}{2}\Big( \frac{(\kappa\mu)^4}{d^2 p}\Big(2 + \frac{r^2}{d}\Big) + \delta\frac{(\kappa\mu)^2 r}{3d^2 p} \Big)^{-1} \Big) \,.$$

When $k = 2$, for rank small $r \leq C\sqrt{d}$, we only need $d^2 p = \Omega(1)$ samples for recovery up to any arbitrary small multiplicative error. When rank $r$ is large, our estimator requires $d^2 p = \Omega(d)$ for both $k \in \{1, 2\}$.

### Acknowledgments

This work was partially supported by NSF grants CNS-1527754, CCF-1553452, CCF-1705007 and GOOGLE Faculty Research Award.

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
