[Supplementary Material · spectrum_supp.pdf]

# Appendix

## A  Algorithm for estimating Schatten $k$-norm for $k \geq 8$

The collection of pseudographs $\mathcal{H}_k$ is partitioned into sets $\{\mathcal{H}_{k,i}^{\mathrm{iso}}\}_{1 \leq i \leq r}$, for some $r \leq k!$. The partitions $\mathcal{H}_{k,i}^{\mathrm{iso}}$ are defined such that the pseudographs in one partition are isomorphic to each other when multi-edges are condensed into one. This is useful since all the pseudographs in one partition are observed together in $G([d], \Omega)$ for any fixed subgraph in $G$. The underlying simple graph (including self loops) for each partition $\mathcal{H}_{k,i}^{\mathrm{iso}}$ is denoted by $F_{k,i}$.

The main idea is to enumerate a list $\mathcal{L}_\ell$ of all connected $\ell$-vertex induced subgraphs (possibly with loops) of the graph $G([d], \Omega)$, for each $1 \leq \ell \leq k$. The unbiased weighted count of all pseudographs $\mathcal{H}_k$ for each of these vertex induced subgraphs $g \in \mathcal{L}_\ell$ is computed. This is achieved by further enumerating a list $\mathcal{S}_{g,\ell}$ of all $\ell$-vertex subgraphs for each $g$. Then the unbiased weight of all pseudographs $H \in \mathcal{H}_k$ that exist in the subgraph $h$ is computed and is summed over to get the estimate of the $k$-th Schatten norm. Recall the notation $\mathcal{P}_\Omega(M)$ which is used to denote the partially observed matrix corresponding to the index set $\Omega$ with the unobserved entries being replaced by zero. We abuse this notation and use $h(M)$ to represent the matrix $M$ restricted to the subgraph $h$ of the observed graph $G([d], \Omega)$.

Each connected induced subgraphs of size $k$ in a graph can be enumerated in time polynomial in $d$ and $k$ [5]. The number of connected induced subgraphs of size $k$ in a graph is upper bounded by $(e\Delta)^k/((\Delta - 1)k)$ where $\Delta$ is the maximum degree of the graph [19]. Therefore, Algorithm 2 runs in time, super exponential in $k$, polynomial in $d$ and the number of $k$ connected induced subgraphs in the observed graph $G([d], \Omega)$.

---

**Algorithm 2** Schatten $k$-norm estimator

---

**Require:** $\mathcal{P}_\Omega(M), k, \mathcal{H}_k, p(H)$ for all $H \in \mathcal{H}_k$
**Ensure:** $\widehat{\Theta}_k(\mathcal{P}_\Omega(M))$
1:  $\widehat{\Theta}_k(\mathcal{P}_\Omega(M)) \leftarrow 0$
2:  **for** $1 \leq \ell \leq k$ **do**
3:     Enumerate a list, $\mathcal{L}_\ell$, of all connected $\ell$-vertex induced subgraphs (possibly with loops) of the graph $G([d], \Omega)$
4:     **for all** $g \in \mathcal{L}_\ell$ **do**
5:        Enumerate a list $\mathcal{S}_{g,\ell}$ of all connected $\ell$-vertex subgraphs of the graph $g$ by removing one or more edges
6:        **for all** $h \in \mathcal{S}_{g,\ell}$ **do**
7:           **for** $1 \leq i \leq r$ **do**
8:              **if** $h$ is isomorphic to $F_{k,i}$ **then**
9:                 $\widehat{\Theta}_k(\mathcal{P}_\Omega(M)) \leftarrow \widehat{\Theta}_k(\mathcal{P}_\Omega(M)) + \sum_{H \in \mathcal{H}_{k,i}^{\mathrm{iso}}} \frac{1}{p(H)} \omega_{h(M)}(H)c(H)$
10:             **end if**
11:          **end for**
12:       **end for**
13:    **end for**
14: **end for**

---

# B From Schatten norms to spectrum and generalized rank

Schatten norms by themselves are rarely of practical interest in real applications, but they provide a popular means to approximate functions of singular values, which are often of great practical interest [4, 20, 13]. In this section, we consider two such applications using the first few Schatten norms explicitly: estimating the generalized rank in Section B.1 and estimating the singular values in Section B.2.

## B.1 Estimating the generalized rank

For a matrix $M \in \mathbb{R}^{d \times d}$ and a given constant $c \geq 0$, its *generalized rank* of order $c$ is given by

$$\text{rank}(M, c) = \sum_{i=1}^{d} \mathbb{I}\big[\sigma_i(M) > c\big]. \tag{11}$$

This recovers the standard rank as a special case when $c = 0$. Without loss of generality, we assume that $\sigma_{\max}(M) \leq 1$. For any given $0 \leq c_2 < c_1 \leq 1$, and $\delta \in [0, 1)$, our goal is to get an estimate $\widehat{r}(\mathcal{P}_\Omega(M))$ from sampled entries $\mathcal{P}_\Omega(M)$ such that

$$(1 - \delta) \, \text{rank}(M, c_1) \quad \leq \quad \widehat{r}(\mathcal{P}_\Omega(M)) \quad \leq \quad (1 + \delta) \, \text{rank}(M, c_2). \tag{12}$$

The reason we take two different constants $c_1, c_2$ is to handle the ambiguous case when the matrix $M$ has many eigenvalues smaller but very close to $c_1$. If we were to set $c_2 = c_1$, then any estimator $\widehat{r}(M)$ would be strictly prohibited from counting these eigenvalues. However, since these eigenvalues are so close to the threshold, distinguishing them from other eigenvalues just above the threshold is difficult. Setting $c_2 < c_1$ allows us to avoid this difficulty and focus on the more fundamental challenges of the problem.

Consider the function $H_{c_1, c_2} : \mathbb{R} \to [0, 1]$ given by

$$H_{c_1, c_2}(x) = \begin{cases} 1 & \text{if } x > c_1 \\ 0 & \text{if } x < c_2 \\ \frac{x - c_2}{c_1 - c_2} & \text{otherwise.} \end{cases} \tag{13}$$

It is a piecewise linear approximation of a step function and satisfies the following:

$$\text{rank}(M, c_1) \quad \leq \quad \sum_{i=1}^{d} H_{c_1, c_2}(\sigma_i(M)) \quad \leq \quad \text{rank}(M, c_2). \tag{14}$$

We exploit this sandwich relation and estimate the generalized rank. Given a polynomial function $f : \mathbb{R} \to \mathbb{R}$ of finite degree $m$ such that $f(x) \approx H_{c_1, c_2}(x)$ for all $x$, such that $f(x) = a_0 + a_1 x + \cdots + a_m x^m$, we immediately have the following relation, which extends to a function on the cone of PSD matrices in the standard way:

$$\sum_{i=1}^{d} f(\sigma_i(M)) \quad = \quad a_0 d + \sum_{k=1}^{m} a_k \|M\|_k^k. \tag{15}$$

Using this equality, we propose the estimator:

$$\widehat{r}(\mathcal{P}_\Omega(M); c_1, c_2) \quad \equiv \quad a_0 d + \sum_{k=1}^{m} a_k \widehat{\Theta}_k(\mathcal{P}_\Omega(M)), \tag{16}$$

where we use the first several $\widehat{\Theta}_k(\mathcal{P}_\Omega(M))$'s obtained by the estimator (3). Note that function $f$ depends upon $c_1, c_2$. The remaining task is to obtain the coefficients of the polynomials in $f$ that is a suitable approximation of the function $H_{c_1, c_2}$. In a similar context of estimating the generalized rank from approximate Schatten norms, [20] propose to use a composite function $f = q_s \circ q$ where $q$ is a finite-degree Chebyshev polynomial of the first kind such that $\sup_{x \in [0,1]} |q(x) - H_{c_1, c_2}(x)| \leq 0.1$, and $q_s$ is a polynomial of degree $2s + 1$ given by

$$q_s(x) \quad = \quad \frac{1}{B(s + 1, s + 1)} \int_0^x t^s (1 - t)^s dt, \qquad \text{where } B(\cdot, \cdot) \text{ is the Beta function.} \tag{17}$$

---

**Algorithm 3** Generalized rank estimator (a variation of [20])

---

**Require:** $\mathcal{P}_\Omega(M), c_1, c_2, s$
**Ensure:** $\widehat{r}(\mathcal{P}_\Omega(M); c_1, c_2)$
 1: For given $c_1$ and $c_2$, find a Chebyshev polynomial of the first kind $q(x)$ satisfying [Appendix C]

$$\sup_{x \in [0,1]} |q(x) - H_{c_1,c_2}(x)| < 0.1$$

 2: Let $C_b$ denote the degree of $q(x)$
 3: Find the degree $(2s+1)C_b$ polynomial expansion of $q_s \circ q(x) = \sum_{k=0}^{(2s+1)C_b} a_k x^k$
 4: $\widehat{r}(\mathcal{P}_\Omega(M); c_1, c_2) \leftarrow a_0 d + \sum_{k=1}^{(2s+1)C_b} a_k \widehat{\Theta}_k(\mathcal{P}_\Omega(M))$    [Algorithm 1]

---

Note that, since $H_{c_1,c_2}$ is a continuous function with bounded variation, classical theory in [17], Theorem 5.7, guarantees existence of the Chebyshev polynomial $q$ of a finite constant degree, say $C_b$, that depends upon $c_1$ and $c_2$. Concretely, for a given choice of thresholds $0 \leq c_1 < c_2 \leq 1$ and degree of the beta approximation $s$, the estimator $\widehat{r}(\mathcal{P}_\Omega(M); c_1, c_2)$ in (16) can be computed as follows.

The approximation of $H_{c_1,c_2}$ with $f = q_s \circ q$ and our upper bound on estimated Schatten norms $\widehat{\Theta}_k(\mathcal{P}_\Omega(M))$ translate into the following guarantee on generalized rank estimator $\widehat{r}(\mathcal{P}_\Omega(M); c_1, c_2)$ given in (16).

**Corollary 2** *Suppose $\|M\|_2 \leq 1$. Under the hypotheses of Theorem 1, for any given $1 \geq c_1 > c_2 \geq 0$, there exists a constant $C_b$, such that for any $s \geq 0$ and any $\gamma > 0$, the estimate in* (16) *with the choice of $f = q_s \circ q$ satisfies*

$$(1-\delta)(\mathrm{rank}(M, c_1) - 2^{-s}d) \leq \widehat{r}(\mathcal{P}_\Omega(M); c_1, c_2) \leq (1+\delta)(\mathrm{rank}(M, c_2) + 2^{-s}d) \quad (18)$$

*with probability at least $1 - \gamma C_b(2s+1)$, where $\delta \equiv \max_{1 \leq k \leq C_b(2s+1)} \left\{ \sqrt{\frac{\rho^2}{\gamma}(\frac{\max\{1, r^{1-2/k}\}}{dp})^k} \right\}$.*

The proof follows immediately using Theorem 1 and the following lemma which gives a uniform bound on the approximation error between $H_{c_1,c_2}$ and $f = q_s \circ q$. Lemma 5, together with Equations. (14) and (15), provides a (deterministic) functional approximation guarantee of

$$\mathrm{rank}(M, c_1) - d\, 2^{-s} \leq \sum_{i=1}^{d} f(\sigma_i(M)) \leq \mathrm{rank}(M, c_1) + d\, 2^{-s}, \quad (19)$$

for any $c_1 < c_2$ and any choice of $s$, as long as $C_b$ is large enough to guarantee 0.1 uniform error bound on the Chebyshev polynomial approximation. Since we can achieve $1 \pm \delta$ approximation on each polynomial in $f(\sigma_i(x))$, Theorem 1 implies the desired Corollary 2. Note that using Remarks 3 and 4, the bounds in (10) hold for $k \in [1, \infty)$ with $r^{1-2/k}$ replaced by $\max\{1, r^{1-2/k}\}$.

**Lemma 5 ([20], Lemma 1)** *Consider the composite polynomial $f(x) = q_s(q(x))$. Then $f(x) \in [0, 1]$ for all $x \in [0, 1]$, and moreover*

$$|f(x) - H_{c_1,c_2}(x)| \leq 2^{-s}, \qquad \text{for all } x \in [0, c_2] \cup [c_1, 1]. \quad (20)$$

In Figure 6, we evaluate the performance of estimator (16) numerically. We construct a symmetric matrix $M$ of size $d = 1000$ and rank $r = 200$. $\sigma_i \sim \mathrm{Uni}(0, 0.4)$ for $1 \leq i \leq r/2$, and $\sigma_i \sim \mathrm{Uni}(0.6, 1)$ for $r/2 + 1 \leq i \leq r$. We estimate $\widehat{r}(\mathcal{P}_\Omega(M); c_1, c_2)$ for Erdös-Rényi sampling $\Omega$, and a choice of $c_2 = 0.5$ and $c_1 = 0.6$, which is motivated by the distribution of $\sigma_i$. We use Chebyshev polynomial of degree $C_b = 2$, and $s = 1$ for $q_s$. That is function $f$ is of degree 6. Accuracy of the estimator can be improved by increasing $C_b$ and $s$, however that would require estimating higher Schatten norms.

## B.2 Estimating the spectrum

Given accurate estimates of first $K$ Schatten norms of a matrix $M$, we can estimate singular values of $M$ using a linear programming based algorithm given in [13]. In particular, we get the following

Figure 6: The left panel shows a histogram of singular values of $M$ chosen for the experiment. The right panel compares absolute error in estimation $\widehat{r}(\mathcal{P}_\Omega(M); c_1 = 0.5, c_2 = 0.6)$ for two choices of the Schatten norm estimates $\widehat{\|M\|_k^k}$: first the proposed estimator $\widehat{\Theta}_k(\mathcal{P}_\Omega(M))$ in (3), and second the Schatten norm of the completed matrix, $\widetilde{M} = \text{AltMin}(\mathcal{P}_\Omega(M))$ from [10].

guarantees on the estimated singular values, whose proof follows directly using the analysis techniques in the proof of [13, Theorem 2]. The main idea is that given the rank, the maximum support size of the true spectrum, and an estimate of its first $K$ moments, one can find $r$ singular values whose $K$ first moments are close to the estimated Schatten norms.

---

**Algorithm 4** Spectrum estimator (a variation of [13])

---

**Require:** $\mathcal{P}_\Omega(M)$, $K$, $\epsilon$, target rank $r$, lower bound $a$ and upper bound $b$ on the positive singular values

**Ensure:** estimated singular values $(\widehat{\sigma}_1, \widehat{\sigma}_2, \ldots, \widehat{\sigma}_r)$

1: $L \in \mathbb{R}^K : L_k = \widehat{\Theta}_k(\mathcal{P}_\Omega(M))$ for $k \in [K]$                 [Algorithm 1]
2: $t = \lceil (b-a)/\epsilon \rceil + 1$, $x \in \mathbb{R}^t : x_i = a + \epsilon(i-1)$, for $i \in [t]$,
3: $V \in \mathbb{R}^{K \times t} : V_{ij} = x_j^i$ for $i \in [K], j \in [t]$
4: $p^* \equiv \{\min_{p \in \mathbb{R}^t} |Vp - L|_1 : \mathbb{1}_t^\top p = 1, p \geq 0\}$
5: $\widehat{\sigma}_i = \min\{x_j : \sum_{\ell \leq j} p_\ell^* \geq \frac{i}{r+1}\}$, $i$th $(r+1)$st-quantile of distribution corresponding to $p^*$

---

Further, our upper bound on the first $K$ moments can be translated into an upper bound on the Wasserstein distance between those two distributions, which in turn gives the following bound on the singular values. With small enough $\epsilon$ and large enough $K$ and $r$, we need sample size $d^2 p > C_{r,K,\epsilon,\gamma} dr^{1-2/k}$ to achieve arbitrary small error.

**Corollary 3** *Under the hypotheses of Theorem 1, given rank $r$, constants $0 \leq a < b$ such that $\sigma_{\min} \geq a$, $\sigma_{\max} \leq b$, and estimates of the first $K$ Schatten norms of $M$, $\{\widehat{\Theta}_k(\mathcal{P}_\Omega(M))\}_{k \in [K]}$ obtained by the estimator (3), for any $0 < \epsilon \ll (b-a)$, and $\gamma > 0$, Algorithm 4 runs in time $\text{poly}(r, K, (b-a)/\epsilon)$ and returns $\{\widehat{\sigma}_i\}_{i \in [r]}$ an estimate of $\{\sigma_i(M)\}_{i \in [r]}$ such that*

$$\frac{1}{r} \sum_{i=1}^r |\widehat{\sigma}_i - \sigma_i| \leq \frac{C(b-a)}{K} + \frac{b-a}{r} + g(K)(b-a)\left(\epsilon K b^{K-1} + \sum_{k=1}^K \sigma_{\max}^k \sqrt{\frac{\rho^2}{\gamma}\left(\frac{\max\{1, r^{1-2/k}\}}{dp}\right)^k}\right),$$
(21)

*with probability at least $1 - \gamma K$, where $C$ is an absolute constant and $g(K)$ only depends on $K$.*

In Figure 7, we evaluate the performance of the proposed estimator (3), in recovering the true spectrum using Algorithm 4. We compare the results with the case when Schatten norms are estimated using matrix completion. We consider two distributions on singular values, one peak and two peaks. More general distributions of spectrum can be recovered accurately, however that would require estimating higher Schatten norms. For both cases, the proposed estimator outperforms matrix completion

approaches, and achieves better accuracy as sample size increases with $\alpha$. In each graph, the black solid line depicts the empirical Cumulative Distribution Function (CDF) of the ground truths $\{\sigma_i\}_{i\in[r]}$ for those $r$ strictly positive singular values. On the left, there are $r$ singular values at one peak $\sigma_i = 1$, and on the right there are $r/2$ singular values at each of the two peaks at $\sigma_i = 1$ and $\sigma_i = 2$. Each blue line and the orange line depicts the empirical CDF of $\{\widehat{\sigma}_i\}_{i\in[d]}$ and $\{\widetilde{\sigma}_i\}_{i\in[d]}$ respectively for each trial, over three independent trials. $\widehat{\sigma}_i$'s are estimated using $\{\widehat{\Theta}_k(\mathcal{P}_\Omega(M))\}_{k\in[K]}$ obtained by the estimator (3), and $\widetilde{\sigma}_i$'s are estimated using $\{\|\widetilde{M}\|_k^k\}_{k\in[K]}$ where $\widetilde{M} = \text{AltMin}(\mathcal{P}_\Omega(M))$, along with Algorithm 2 in [13], for $K = 7$. $M$ is a symmetric matrix of size $d = 1000$ and rank $r \in \{50, 200, 500\}$ with singular values $\{\sigma_i\}_{i\in[d]}$. $\Omega$ is generated using Erdös-Rényi sampling with probability $p = (\alpha/d)r^{1-2/7}$ for $\alpha \in \{3, 5, 8, 10\}$.

Figure 7: The proposed estimator (in blue solid lines) outperforms matrix completion approaches (in orange solid lines) in estimating the ground truths empirical cumulative distribution function of the $r$ strictly positive singular values (in black solid line) for two examples: one peak at $\sigma_i = 1$ on the left and two peaks at $\sigma_i = 1$ or $\sigma_i = 2$ on the right. Both approaches achieve better accuracy as sample size increases with $\alpha$, where $p = (\alpha/d)r^{1-2/7}$.

# C   Algorithm for computing the Chebyshev polynomial

---

**Algorithm 5** Chebyshev polynomial of the first kind approximating $H_{c_1,c_2}(x)$

---

**Require:** $H_{c_1,c_2}$, $c_1$, $c_2$, and target accuracy $\delta = 0.1$
**Ensure:** Chebyshev polynomial $q(x)$ of first kind
1: $g(x) \equiv \frac{x-c_2}{c_1-c_2}$
2: $T_0(x) \equiv 1, T_1(x) \equiv x$
3: $q(x) \leftarrow \frac{1}{\pi}\int_{c_2}^{c_1}(1-x^2)^{-1/2}g(x)T_0(x)dx + \frac{1}{\pi}\int_{c_1}^{1}(1-x^2)^{-1/2}T_0(x)dx$
4: $i = 1$
5: **while** $\sup_{x\in[0,1]}|q(x) - H_{c_1,c_2}(x)| \geq \delta$ **do**
6: $\quad q(x) \leftarrow q(x) + \frac{2T_i(x)}{\pi}\int_{c_2}^{c_1}(1-x^2)^{-1/2}g(x)T_i(x)dx + \frac{2T_i(x)}{\pi}\int_{c_1}^{1}(1-x^2)^{-1/2}T_i(x)dx$
7: $\quad i \leftarrow i + 1$
8: $\quad T_i(x) \equiv 2xT_{i-1}(x) - T_{i-2}(x)$
9: **end while**

---

# D Proofs

We provide proofs for main results and technical lemmas.

## D.1 Proof of Theorem 1

Consider $\widetilde{W}$ to be the collection of all length $k$ closed walks on a complete graph of $d$ vertices. Here we slightly overload the notion of complete graph to refer to an undirected graph with not only all the $d(d-1)/2$ simple edges but also with $d$ self loops as well. Construct the largest possible collection $W$ from $\widetilde{W}$ wherein each walk has distinct weights that is $\omega(w) \neq \omega(w')$ for all $w, w' \in W$. We partition $W$ according to the pattern among $k$-cyclic pseudographs, which are further partitioned into four groups. The estimator (3) can be re-written as

$$
\begin{aligned}
\widehat{\Theta}_k(\mathcal{P}_\Omega(M)) &= \sum_{w \in W} \frac{c(H(w))}{p(H(w))} \, \omega_{\mathcal{P}_\Omega(M)}(w) \\
&= \sum_{H \in \mathcal{H}_k} \left\{ \frac{c(H)}{p(H)} \sum_{w:H(w)=H} \omega_M(w) \, \mathbb{I}(w \subseteq \Omega) \right\} \qquad (22) \\
&= \sum_{i=1}^{4} \sum_{H \in \mathcal{H}_{k,i}} \left\{ \frac{c(H)}{p(H)} \sum_{w:H(w)=H} \omega_M(w) \, \mathbb{I}(w \subseteq \Omega) \right\}, \qquad (23)
\end{aligned}
$$

where we write $w \subseteq \Omega$ to denote the event that all the edges in the walk $w$ are sampled, and we define

- $\mathcal{H}_{k,1} \equiv \{C_k\}$ is just a (set of a) simple cycle of length $k$ and there are total $|\{w \in W : H(w) \in \mathcal{H}_{k,1}\}| = \binom{d}{k}(k!/2k) \leq (d^k/2k)$ corresponding walks to this set, and $c(C_k) = 2k$.

- $\mathcal{H}_{k,2} \equiv \{H(V_H, E_H) \in \mathcal{H}_k : |V_H| \leq k-1 \text{ and no self loops}\}$, and there are total $|\{w \in W : H(w) \in \mathcal{H}_{k,2}| \leq d^{k-1}$ corresponding walks to this set.

- $\mathcal{H}_{k,3} \equiv \bigcup_{s=1}^{k-1} \mathcal{H}_{k,3,s}$ where $\mathcal{H}_{k,3,s} = \{H \in \mathcal{H}_k \text{ with } s \text{ self loops}\}$, and there are total $|\{w \in W : H(w) \in \mathcal{H}_{k,3}\}| \leq d^{k-s}$ corresponding walks in this set.

- $\mathcal{H}_{k,4} \equiv \{H(V_H, E_H) \in \mathcal{H}_k : |V_H| = 1\}$ is a (set of a) graph with $k$ self loops and there are total $|\{w \in W : H(w) \in \mathcal{H}_{k,4}\}| = d$ corresponding walks to this set.

Given this unbiased estimator, we provide an upper bound on the variance of each of the partitions to prove concentration with Chebyshev's inequality. For any walk $w \in W$, let $|w|$ denote the number of unique edges (including self loops) that the walk $w$ traverses. Let $|w \cap w'|$ denote the number of unique overlapping edges (including self loops) of walks $w$ and $w'$. We have,

$$
\begin{aligned}
\mathrm{Var}\big(\widehat{\Theta}_k(\mathcal{P}_\Omega(M))\big) &= 2\sum_{\ell=1}^{k-1} \sum_{\substack{w \neq w' \in \widetilde{W} \\ |w \cap w'|=\ell}} \mathrm{Covar}\left( \frac{\mathbb{I}(w \subseteq \Omega)\omega_M(w)c(H(w))}{p(H(w))}, \frac{\mathbb{I}(w' \subseteq \Omega)\omega_M(w')c(H(w'))}{p(H(w'))} \right) \\
&\quad + \sum_{i=1}^{4} \sum_{H \in \mathcal{H}_{k,i}} \left\{ \frac{c(H)^2}{p(H)^2} \sum_{w:H(w)=H} \omega_M(w)^2 \mathrm{Var}\big(\mathbb{I}(w \subseteq \Omega)\big) \right\} \qquad (24) \\
&< 4\sum_{\ell=1}^{k-1} \sum_{\substack{w \neq w' \in W \\ |w \cap w'|=\ell}} \mathbb{E}\Big[ \mathbb{I}(w \subseteq \Omega)\mathbb{I}(w' \subseteq \Omega) \Big] \left( \frac{\big|\omega_M(w)\,\omega_M(w')\big|c(H(w))c(H(w'))}{p(H(w))\,p(H(w'))} \right) \\
&\quad + \sum_{i=1}^{4} \sum_{H \in \mathcal{H}_{k,i}} \sum_{w:H(w)=H} \frac{c(H)^2 \omega_M(w)^2}{p(H)^2} \mathbb{E}\Big[ \mathbb{I}(w \subseteq \Omega) \Big]. \qquad (25)
\end{aligned}
$$

Recall from the definition of incoherence that $|M_{ii}| \leq \sigma_1(M)\mu r/d$ and $|M_{ij}| = \sigma_1(M)\mu r^{1/2}/d$, and let $\alpha = \sigma_1(M)\mu r^{1/2}/d$ denote the maximum off-diagonal entry, such that $|M_{ij}| \leq \alpha$ and

$|M_{ii}| \leq \alpha\sqrt{r}$ for all $i,j \in [d]$. Let $A_{p,k,\alpha,d} = d^k\alpha^{2k}/p^k$ denote the target scaling of the variance, then

$$\sum_{H \in \mathcal{H}_{k,i}} \sum_{w:H(w)=H} \frac{c(H)^2\,\omega_M(w)^2}{p(H)^2} \mathbb{E}\Big[\mathbb{I}(w \subseteq \Omega)\Big] \leq$$

$$\begin{cases}
\dfrac{d^k}{2k}\dfrac{(2k)^2\alpha^{2k}}{p^k} = 2kA_{p,k,\alpha,d}\,, & \text{for } i = 1\ , & (26)\\[3ex]
d^{k-1}\dfrac{f(k)^2\alpha^{2k}}{p^k} = \dfrac{f(k)^2}{d}A_{p,k,\alpha,d}\,, & \text{for } i = 2\ , & (27)\\[3ex]
d\dfrac{r^k\alpha^{2k}}{p} = \dfrac{r^k p^{k-1}}{d^{k-1}}A_{p,k,\alpha,d}\,, & \text{for } i = 4\ , & (28)
\end{cases}$$

and for $i = 3$ and for $1 \leq s \leq k-1$, we have

$$\sum_{H \in \mathcal{H}_{k,3,s}} \sum_{w:H(w)=H} \frac{c(H)^2\,\omega_M(w)^2}{p(H)^2} \mathbb{E}\Big[\mathbb{I}(w \subseteq \Omega)\Big] \leq d^{k-s}\frac{f(k)^2\alpha^{2k}r^s}{p^k} = \frac{f(k)^2 r^s}{d^s}A_{p,k,\alpha,d}\ , \quad (29)$$

where $c(H)$ is defined as the multiplicity of walks with the same weight satisfying $c(H) \leq f(k)$. For $w \neq w'$ and $|w \cap w'| = \ell$, where the range of $\ell$ varies across equations depending upon the set to which $w, w'$ belongs, we have the following:

$$\sum_{\substack{w \neq w' \in W \\ |w \cap w'|=\ell, H(w) \in \mathcal{H}_{k,i,s}, H(w') \in \mathcal{H}_{k,i',s'}}} \mathbb{E}\Big[\mathbb{I}(w \in \Omega)\mathbb{I}(w' \in \Omega)\Big] \frac{\big|\omega_M(H(w))\omega_M(H(w'))\big|\,c(H(w))c(H(w'))}{p(H(w))p(H(w'))} \leq$$

$$\begin{cases}
\dfrac{d^k d^{k-(\ell+1)}}{2k}\dfrac{\alpha^{2k}(2k)^2}{p^\ell} = \dfrac{(dp)^{k-\ell}}{d}2kA_{p,k,\alpha,d}, & \text{for } i=i'=1\ (30)\\[3ex]
\dfrac{f(k)^2 d^{k-1}d^{k-1-(\ell+1)}\alpha^{2k}}{p^\ell} \leq \dfrac{f(k)^2(dp)^{k-\ell}}{d^3}A_{p,k,\alpha,d} & \text{for } i=i'=2\ (31)\\[3ex]
\dfrac{f(k)^2 d^{k-s}d^{k-s'-\ell}\alpha^{2k-s-s'}(\alpha\sqrt{r})^{s+s'}}{p^\ell} \leq \dfrac{f(k)^2(dp)^{k-\ell}}{(d/\sqrt{r})^{s+s'}}A_{p,k,\alpha,d}\,, & \text{for } i=i'=3\ (32)\\[3ex]
\dfrac{f(k)^2 d^k d^{k-1-(\ell+1)}\alpha^{2k}}{p^\ell} \leq \dfrac{f(k)^2(dp)^{k-\ell}}{d^2}A_{p,k,\alpha,d} & \text{for } i=1, i'=2\ (33)\\[3ex]
\dfrac{f(k)^2 d^k d^{k-s-(\ell+1)}\alpha^{2k-s}(\alpha\sqrt{r})^s}{p^\ell} \leq \dfrac{f(k)^2(dp)^{k-\ell}}{d(d/\sqrt{r})^s}A_{p,k,\alpha,d} & \text{for } i=1, i'=3\ (34)\\[3ex]
\dfrac{f(k)^2 d^{k-1} d^{k-s-(\ell+1)}\alpha^{2k-s}(\alpha\sqrt{r})^s}{p^\ell} \leq \dfrac{f(k)^2(dp)^{k-\ell}}{d^2(d/\sqrt{r})^s}A_{p,k,\alpha,d} & \text{for } i=2, i'=3\ (35)\\[3ex]
\dfrac{f(k)^2 dd^{k-s-\ell}\alpha^{k-s}(\alpha\sqrt{r})^{k+s}}{p^\ell} \leq \dfrac{f(k)^2(dp)^{k-\ell}}{d^{k-1}(d/\sqrt{r})^{k+s}}A_{p,k,\alpha,d} & \text{for } i=3, i'=4\ (36)
\end{cases}$$

where (36) is valid only for $\ell = 1$. Note that for any $w$ with $H(w) \in \mathcal{H}_{k,1} \bigcup \mathcal{H}_{k,2}$, it has no overlap with $w'$ such that $H(w') \in \mathcal{H}_{k,4}$.

Observe that $\mathrm{Var}\big(\widehat{\Theta}_k(\mathcal{P}_\Omega(M))\big)$ as bounded in (25) is upper bounded by the sum of quantities in (26)-(36), summating over all possible values of $1 \leq \ell \leq k-1$, and $1 \leq s, s' \leq k-1$. Let $h(k) \equiv f(k)^2 A_{p,k,\alpha,d}$. Observe that quantities in (26),(27), and (29) are upper bounded by $h(k)$. Quantities in (30)-(36) are upper bounded by $h_1(k) \equiv h(k)(dp)^{k-1}/d$. Quantity in (28) is upper bounded by $h_2(k) \equiv h(k)r^k p^{k-1}/d^{k-1}$.

Given $\|M\|_k^k \geq r(\sigma_{\min})^k$, recall a bound on off diagonals of matrix $M$ by $|M_{ij}| \leq \alpha = \mu\sigma_{\max}\sqrt{r}/d$ and $A_{p,k,\alpha,d} = d^k\alpha^{2k}/p^k$. This gives

$$\frac{A_{p,k,\alpha,d}}{\|M\|_k^{2k}} \leq \frac{\kappa^{2k}\mu^{2k}r^{k-2}}{d^k p^k}\,. \tag{37}$$

Using Chebyshev's inequality and collecting all terms in the upper bound on the variance, we have for sufficiently large $d$, the following bound:

$$\mathbb{P}\left(\frac{\left|\widehat{\Theta}_k(\mathcal{P}_\Omega(M)) - \|M\|_k^k\right|}{\|M\|_k^k} \geq \delta\right) \leq \frac{(\kappa\mu)^{2k}f(k)^2r^{k-2}}{\delta^2(dp)^k}\max\left\{1, \frac{(dp)^{k-1}}{d}, \frac{r^kp^{k-1}}{d^{k-1}}\right\} \quad (38)$$

where the second and the third term in the $\max$ expression follow by evaluating $h_1(k)$ and $h_2(k)$. If sampling probability $p$ is small enough such that $dp \leq Cd^{1/(k-1)}$ for some constant $C$, then the second and the third terms are smaller than the first term. Hence, the desired result in Theorem 1 follows.

## D.2 Proof of Theorem 2

We can prove a Bernstien-type bound on accuracy of the estimator. The estimator (3) can be re-written as a multi-linear polynomial function of $d(d+1)/2$ i.i.d. Bernoulli($p$) random variables.

$$\widehat{\Theta}_k(\mathcal{P}_\Omega(M)) = \sum_{w \in W}\left\{\frac{c(H(w))}{p(H(w))}\omega_M(w)\prod_{(i,j)\in\text{unique}(w)}\mathbb{I}((i,j)\in\Omega)\right\}, \quad (39)$$

where $\mathbb{I}((i,j)\subseteq\Omega)$ is a random variable that takes value 1 if the $(i,j)_{\text{th}}$ entry of the matrix $M$ is sampled, and $\text{unique}(w)$ denotes the set of the unique edges (and self loops) that the walk $w$ traverses. Let $q$ denote the power of the polynomial function that is the maximum number of unique edges in the walk $w$, that is $q = k$.

We use the following Bernstien-type concentration results of [18] for the polynomials of independent random variables.

**Lemma 6 ([18],Theorem 1.3)** *We are given $d(d+1)/2$ independent central moment bounded random variables $\{\mathbb{I}((i,j)\in\Omega)\}_{1\leq i\leq j\leq d}$ with same parameter $L$. We are given a multilinear polynomial $\widehat{\Theta}_k(\mathcal{P}_\Omega(M))$ of power $q$, then*

$$\mathbb{P}\left[\left|\widehat{\Theta}_k(\mathcal{P}_\Omega(M)) - \mathbb{E}\left[\widehat{\Theta}_k(\mathcal{P}_\Omega(M))\right]\right| \geq \lambda\right] \leq e^2\max\left\{e^{\frac{-\lambda^2}{\text{Var}[\widehat{\Theta}_k(\mathcal{P}_\Omega(M))]R^q}}, \max_{t\in[q]}e^{-\left(\frac{\lambda}{\mu_t L^t R^q}\right)^{1/t}}\right\} \quad (40)$$

*where $R$ is some absolute constant and $\mu_t$ is defined as follows:*

$$\mu_t = \max_{\substack{S\subseteq\{(i,j):i,j\in[d]\} \\ |S|=t}}\left(\sum_{w\in W|w\supseteq S}\frac{c(H(w))}{p(H(w))}|\omega_M(w)|\prod_{(i,j)\in\text{unique}(w)\setminus S}\mathbb{E}[\mathbb{I}((i,j)\in\Omega)]\right) \quad (41)$$

*where $w \supseteq S$ denotes that the walk $w$ comprises edges(and self loops) contained in the set $S$. $L$ is defined as follows: A random variable $Z$ is called central moment bounded with real parameter $L > 0$, if for any integer $i \geq 1$ we have*

$$\mathbb{E}\left[|Z - \mathbb{E}[Z]|^i\right] \leq iL\mathbb{E}[|Z - \mathbb{E}[Z]|^{i-1}]. \quad (42)$$

For Bernoulli random variables $L \in [1/4, 1]$. In the following, we show that $\mu_t \leq (\mu\sigma_{\max})^k g(k)r^k/(d(dp)^t)$, for $t \in [k]$. Using Lemma 6, along with $\|M\|_k^k \geq r(\sigma_{\min})^k$, the bound in (10) follows immediately.

To compute $\mu_t$, define a set of walks $W_{\ell,s,\hat{s}}$ such that $w \in W_{\ell,s,\hat{s}}$ has $0 \leq \ell \leq k$ unique edges and $0 \leq s \leq k$ unique self loops, and $\hat{s}$ total self loops with $\ell + \hat{s} \leq k$. For the set $S$ as required in (41), let $S_{\tilde{\ell},\tilde{s}}$ be a set of $\tilde{\ell}$ unique edges and $\tilde{s}$ unique self loops, with $|S_{\tilde{\ell},\tilde{s}}| = \tilde{\ell} + \tilde{s}$ where $1 \leq \tilde{\ell} + \tilde{s} \leq k$.

Therefore, we have

$$
\begin{aligned}
\mu_t &= \max_{\substack{S_{\tilde{\ell},\tilde{s}} \\ :\tilde{\ell}+\tilde{s}=t}} \left( \sum_{\substack{0\leq s\leq \hat{s}\leq k \\ \ell\in[k]:\ell+\hat{s}\leq k}} \sum_{\substack{w\in W_{\ell,s,\hat{s}} \\ :w\supseteq S_{\tilde{\ell},\tilde{s}}}} \frac{c(H(w))}{p(H(w))} |\omega_M(w)| \prod_{(i,j)\in \mathrm{unique}(w)\setminus S_{\tilde{\ell},\tilde{s}}} \mathbb{E}[\mathbb{I}((i,j)\subseteq \Omega)] \right) \\
&\leq \max_{\substack{S_{\tilde{\ell},\tilde{s}} \\ :\tilde{\ell}+\tilde{s}=t}} \left( \sum_{\substack{0\leq s\leq \hat{s}\leq k \\ \ell\in[k]:\ell+\hat{s}\leq k}} \sum_{\substack{w\in W_{\ell,s,\hat{s}} \\ :w\supseteq S_{\tilde{\ell},\tilde{s}}}} \frac{f(k)}{p^{\ell+s}} \alpha^k r^{\hat{s}/2} p^{\ell+s-(\tilde{\ell}+\tilde{s})} \right) \\
&\leq \max_{\substack{S_{\tilde{\ell},\tilde{s}} \\ :\tilde{\ell}+\tilde{s}=t}} \left( \sum_{\substack{0\leq s\leq \hat{s}\leq k \\ \ell\in[k]:\ell+\hat{s}\leq k,\tilde{s}\leq s}} \frac{d^{\ell-(1+\tilde{\ell})}f(k)}{p^{\ell+s}} \frac{(\mu\sigma_{\max})^k r^{(k+\hat{s})/2}}{d^k} p^{\ell+s-(\tilde{\ell}+\tilde{s})} \right) \\
&= \max_{\substack{S_{\tilde{\ell},\tilde{s}} \\ :\tilde{\ell}+\tilde{s}=t}} \left( \sum_{\substack{0\leq s\leq \hat{s}\leq k \\ \ell\in[k]:\ell+\hat{s}\leq k,\tilde{s}\leq s}} \frac{f(k)(\mu\sigma_{\max})^k r^{(k+\hat{s})/2}}{dd^{(k-\ell-\tilde{s})}(dp)^{(\tilde{\ell}+\tilde{s})}} \right) \\
&\leq \max_{\substack{S_{\tilde{\ell},\tilde{s}} \\ :\tilde{\ell}+\tilde{s}=t}} \left( \frac{k^3 f(k)(\mu\sigma_{\max})^k r^{(k+\hat{s})/2}}{dd^{(k-\ell-\tilde{s})}(dp)^{(\tilde{\ell}+\tilde{s})}} \right) \leq \frac{(\mu\sigma_{\max})^k g(k) r^k}{d(dp)^t}.
\end{aligned}
$$

Figure 8: The 4-cyclic pseudographs $\mathcal{H}_4$.

Figure 9: The 5-cyclic pseudographs $\mathcal{H}_5$.

# E  $k$-cyclic pseudographs

Figure 10: The 6-cyclic pseudographs $\mathcal{H}_6$.

$E_1$     $E_2$     $E_3$     $E_4$

$E_5$     $E_6$     $E_7$     $E_8$

$E_9$     $E_{10}$     $E_{11}$     $E_{12}$

$E_{13}$     $E_{14}$     $E_{15}$     $E_{16}$

$E_{17}$     $E_{18}$     $E_{19}$     $E_{20}$

$E_{21}$     $E_{22}$     $E_{23}$     $E_{24}$

$E_{25}$     $E_{26}$     $E_{27}$     $E_{28}$

$E_{29}$     $E_{30}$     $E_{31}$     $E_{32}$

Figure 11: The 7-cyclic pseudographs $\mathcal{H}_7$

$E_{33}$

$E_{34}$

$E_{35}$

$E_{36}$

$E_{37}$

$E_{38}$

$E_{39}$

$E_{40}$

$E_{41}$

$E_{42}$

$E_{43}$

$E_{44}$

$E_{45}$

$E_{46}$

$E_{47}$

$E_{48}$

$E_{49}$

$E_{50}$

$E_{51}$

$E_{52}$

$E_{53}$

$E_{54}$

$E_{55}$

$E_{56}$

$E_{57}$

$E_{58}$

$E_{59}$

$E_{60}$

Figure 12: The 7-cyclic pseudographs $\mathcal{H}_7$

Figure 13: The 7-cyclic pseudographs $\mathcal{H}_7$.

# F    Efficient computation of $\omega_M(H)$ for $k \in \{4, 5, 6, 7\}$

In this section we provide the complete matrix oeprations for copmuting $\gamma_M(H)$'s. Equations (43) - (49) give expressions to compute $\gamma_M(H)$ for $H \in \mathcal{H}_4$ as labeled in Figure 8. Equations (50) - (61) give expressions to compute $\gamma_M(H)$ for $H \in \mathcal{H}_5$ as labeled in Figure 9. Equations (62) - (93) give expressions to compute $\gamma_M(H)$ for $H \in \mathcal{H}_6$ as labeled in Figure 10. Equations (94) - (186) give expressions to compute $\gamma_M(H)$ for $H \in \mathcal{H}_7$ as labeled in Figure 13.

For brevity of notations and readability, we define the following additional notations. Let $A \odot B$ denote the Hadamard product. For $A \in \mathbb{R}^{d \times d}$, let $\mathrm{sum}(A)$ denote a vector $v \in \mathbb{R}^d$ such that $v_i = \sum_{j \in [d]} A_{i,j}$. With a slight abuse of notation, for $v \in \mathbb{R}^d$, let $\mathrm{sum}(v)$ denote sum of all elements of $v$ that is $\mathrm{sum}(v) = \sum_{i \in [d]} v_i$. Let $\mathrm{sum}(\gamma_M(H_i) : \gamma_M(H_j)) \equiv \sum_{i'=i}^{j} \gamma_M(H_{i'})$. Define $R \equiv \mathbb{1}_{d \times d} - \mathrm{diag}(\mathbb{1}_{d \times d})$, that is $R$ is an all-ones matrix except on diagonals which are zeros. Further, for brevity, we omit the subscript $M$ from the notations $\gamma_M(H), O_M$ and $D_M$.

$$\gamma(B_1) = \text{sum}(\text{sum}(D \odot D \odot D \odot D)) \tag{43}$$

$$\gamma(B_2) = \text{sum}(\text{sum}(O \odot O \odot O \odot O)) \tag{44}$$

$$\gamma(B_3) = 4 \, \text{tr}(O*O*D*D) \tag{45}$$

$$\gamma(B_4) = 2 \, \text{sum}(\text{sum}((O \odot O)*(O \odot O) \odot R)) \tag{46}$$

$$\gamma(B_5) = 2 \, \text{tr}(O*D*O*D) \tag{47}$$

$$\gamma(B_6) = \text{tr}(O*O*O*O) - \text{sum}(\gamma(B_2):\gamma(B_4)) \tag{48}$$

$$\gamma(B_7) = \text{tr}(M*M*M*M) - \text{sum}(\gamma(B_1):\gamma(B_6)) \tag{49}$$

$$\gamma(C_1) = \text{tr}(D \odot D \odot D \odot D \odot D) \tag{50}$$

$$\gamma(C_2) = 5 \, \text{sum}(\text{sum}(D*O \odot O \odot O \odot O)) \tag{51}$$

$$\gamma(C_3) = 5 \, \text{sum}(\text{sum}((D \odot D \odot D)*(O \odot O))) \tag{52}$$

$$\gamma(C_4) = 5 \, \text{tr}((O \odot O \odot O)*O*O) \tag{53}$$

$$\gamma(C_5) = 5 \, \text{sum}(\text{sum}(D*(O \odot O)*(D \odot D))) \tag{54}$$

$$\gamma(C_6) = 5 \, \text{sum}(\text{sum}(((O \odot O)*D*(O \odot O)) \odot R)) \tag{55}$$

$$\gamma(C_7) = 5 \, \text{sum}(\text{sum}((D*(O \odot O)*(O \odot O)) \odot R)) \tag{56}$$

$$\gamma(C_8) = 5 \, \text{tr}(O*O*O*(D \odot D)) \tag{57}$$

$$\gamma(C_9) = 5 \, \text{sum}(\text{diag}(O \odot O \odot O) \odot \text{sum}(O \odot O)) - 10 \, \text{tr}((O \odot O \odot O)*O*O)) \tag{58}$$

$$\gamma(C_{10}) = \text{tr}(O*O*O*O*O) - \gamma(C_4) - \gamma(C_9) \tag{59}$$

$$\gamma(C_{11}) = 5 \, \text{tr}(O*D*O*D*O) \tag{60}$$

$$\gamma(C_{12}) = \text{tr}(M*M*M*M*M) - \text{sum}(\gamma(C_1):\gamma(C_{11})) \tag{61}$$

$$\gamma(D_1) = \text{sum}(\text{sum}(D \odot D \odot D \odot D \odot D \odot D)) \tag{62}$$

$$\gamma(D_2) = \text{sum}(\text{sum}(O \odot O \odot O \odot O \odot O \odot O)) \tag{63}$$

$$\gamma(D_3) = 6\,\text{sum}(\text{sum}(((O \odot O) * (O \odot O \odot O \odot O)) \odot R)) \tag{64}$$

$$\gamma(D_4) = 6\,\text{sum}(\text{sum}(((O \odot O) * (D \odot D \odot D \odot D)) \odot R)) \tag{65}$$

$$\gamma(D_5) = 9\,\text{sum}(\text{sum}(((D \odot D) * (O \odot O \odot O \odot O)) \odot R)) \tag{66}$$

$$\gamma(D_6) = 3\,\text{sum}(\text{sum}(((D \odot D) * (O \odot O) * (D \odot D)) \odot R)) \tag{67}$$

$$\gamma(D_7) = 6\,\text{sum}(\text{sum}(((D \odot D) * (O \odot O) * (O \odot O)) \odot R)) \tag{68}$$

$$\gamma(D_8) = 9\,\text{sum}(\text{sum}(((O \odot O) * (D \odot D) * (O \odot O)) \odot R)) \tag{69}$$

$$\gamma(D_9) = 6\,\text{sum}(\text{sum}(((D \odot D \odot D) * (O \odot O) * D) \odot R)) \tag{70}$$

$$\gamma(D_{10}) = 6\,\text{sum}(\text{sum}((D * (O \odot O \odot O \odot O) * D) \odot R)) \tag{71}$$

$$\gamma(D_{11}) = 3\,\text{sum}\Big(\big(\text{sum}(((O \odot O) * (O \odot O)) \odot R)\big) \odot \big(\text{sum}(O \odot O)\big) - \text{sum}\Big(((O \odot O \odot O \odot O) * (O \odot O)) \odot R\Big)$$
$$- \text{diag}((O \odot O) * (O \odot O) * (O \odot O))\Big) \tag{72}$$

$$\gamma(D_{12}) = 4\,\text{tr}((O \odot O) * (O \odot O) * (O \odot O)) \tag{73}$$

$$\gamma(D_{13}) = 2\,\text{sum}\Big(\big(\text{sum}(O \odot O)\big) \odot \big(\text{sum}(O \odot O)\big) \odot \big(\text{sum}(O \odot O)\big) - \text{sum}((O \odot O \odot O \odot O \odot O \odot O))$$
$$- 3\left(\big(\text{sum}(O \odot O \odot O \odot O)\big) \odot \big(\text{sum}(O \odot O)\big) - \big(\text{sum}(O \odot O \odot O \odot O \odot O \odot O)\big)\right)\Big) \tag{74}$$

$$\gamma(D_{14}) = 3\,\text{sum}(\text{sum}((D * (O \odot O) * (O \odot O) * D) \odot R)) \tag{75}$$

$$\gamma(D_{15}) = 12\,\text{sum}(\text{sum}((D * (O \odot O) * D * (O \odot O)) \odot R)) \tag{76}$$

$$\gamma(D_{16}) = 6\,\text{sum}\Big(\text{sum}(((O \odot O \odot O) * O) \odot R \odot (O * O)) - \text{sum}(((O \odot O \odot O \odot O) * (O \odot O)) \odot R)\Big) \tag{77}$$

$$\gamma(D_{17}) = 6\,\text{tr}((D \odot D \odot D) * O * O * O) \tag{78}$$

$$\gamma(D_{18}) = 24\,\text{tr}(D * (O \odot O \odot O) * O * O) \tag{79}$$

$$\gamma(D_{19}) = 6\,\text{tr}(D * O * (O \odot O \odot O) * O) \tag{80}$$

$$\gamma(D_{20}) = 6\left(\text{sum}(\text{sum}((O * O) \odot ((O * (D \odot D) * O) \odot R))) - \text{sum}(\text{sum}(((O \odot O) * (D \odot D) * (O \odot O)) \odot R))\right) \tag{81}$$

$$\gamma(D_{21}) = 12\,\text{tr}(O * (D \odot D) * O * D * O) \tag{82}$$

$$\gamma(D_{22}) = 6\left(\text{sum}\Big(\text{sum}\big(((O * O) \odot R \odot (O * O) - ((O \odot O) * (O \odot O)) \odot R)\big) \odot \text{sum}(O \odot O)\Big)\right.$$
$$- 2\,\text{sum}\Big(\text{sum}\big((((O \odot O \odot O) * O) \odot R \odot (O * O) - ((O \odot O \odot O \odot O) * (O \odot O)) \odot R)\big)\Big)$$
$$\left.- \text{sum}\Big(\text{sum}\big(((O * O) \odot R \odot (O * O) - ((O \odot O) * (O \odot O)) \odot R) \odot (O \odot O)\big)\Big)\right) \tag{83}$$

$$\gamma(D_{23}) = 9\,\text{sum}(\text{sum}(((O * O) \odot R \odot (O * O) - ((O \odot O) * (O \odot O)) \odot R) \odot ((O \odot O)))) \tag{84}$$

$$\gamma(D_{24}) = 12\,\text{sum}(\text{diag}(O * D * O * O) \odot \text{sum}((O \odot O)) - \text{diag}((O \odot O \odot O) * D * O * O)$$
$$- \text{diag}((O \odot O \odot O) * O * D * O)) \tag{85}$$

$$\gamma(D_{25}) = 6\,\text{sum}(\text{diag}(O * O * O) \odot \text{sum}((O \odot O) * D) - 2\,\text{diag}((O \odot O \odot O) * D * O * O)) \tag{86}$$

$$\gamma(D_{26}) = 12\,\text{sum}(\text{diag}(O * O * O) \odot \text{diag}(D) \odot \text{sum}((O \odot O)) - \text{diag}((O \odot O \odot O) * O * O) \odot \text{diag}(D)) \tag{87}$$

$$\gamma(D_{27}) = 3\,\text{sum}\Big(\text{diag}(O * O * O) \odot \text{diag}(O * O * O) - 2\,\text{diag}((O \odot O) * (O \odot O) * (O \odot O))\Big)$$
$$- (4/3)\gamma(D_{23}) \tag{88}$$

$$\gamma(D_{28}) = \mathrm{tr}(O*O*O*O*O*O) - \gamma(D_2) - \gamma(D_3) - \gamma(D_{11}) - \gamma(D_{12}) - \gamma(D_{13})$$
$$-\gamma(D_{16}) - \gamma(D_{22}) - \gamma(D_{23}) - \gamma(D_{27}) \tag{89}$$

$$\gamma(D_{29}) = 2\,\mathrm{tr}(D*O*D*O*D*O) \tag{90}$$

$$\gamma(D_{30}) = 3\,\mathrm{sum}(\mathrm{sum}((O*D*O)\odot R\odot(O*D*O)) - \mathrm{sum}(((O\odot O)*(D\odot D)*(O\odot O))\odot R)) \tag{91}$$

$$\gamma(D_{31}) = 6\,\mathrm{sum}(\mathrm{sum}((O*D*O*D)\odot R\odot(O*O)) - \mathrm{sum}(((O\odot O)*D*(O\odot O)*D)\odot R)) \tag{92}$$

$$\gamma(D_{32}) = \mathrm{tr}(M*M*M*M*M*M) - \mathrm{tr}(O*O*O*O*O*O) - \mathrm{sum}(\gamma(D_1) : \gamma(D_{26})) + \gamma(D_2) + \gamma(D_3) +$$
$$\gamma(D_{11}) + \gamma(D_{12}) + \gamma(D_{13}) + \gamma(D_{16}) + \gamma(D_{22}) + \gamma(D_{23}) - \gamma(D_{29}) - \gamma(D_{30}) - \gamma(D_{31}) \tag{93}$$

$$\gamma(E_1) = \mathrm{sum}(\mathrm{diag}((D\odot D\odot D\odot D\odot D\odot D\odot D))) \tag{94}$$

$$\gamma(E_2) = 7\,\mathrm{sum}(\mathrm{sum}((O\odot O)*(D\odot D\odot D\odot D\odot D))) \tag{95}$$

$$\gamma(E_3) = 7\,\mathrm{sum}(\mathrm{sum}(((D\odot D)*(O\odot O)*(D\odot D\odot D))\odot R)) \tag{96}$$

$$\gamma(E_4) = 14\,\mathrm{sum}(\mathrm{sum}((O\odot O\odot O\odot O)*(D\odot D\odot D))) \tag{97}$$

$$\gamma(E_5) = 7\,\mathrm{sum}(\mathrm{sum}((O\odot O\odot O\odot O\odot O\odot O)*D)) \tag{98}$$

$$\gamma(E_6) = 7\,\mathrm{sum}(\mathrm{sum}((D*(O\odot O)*(D\odot D\odot D\odot D))\odot R)) \tag{99}$$

$$\gamma(E_7) = 21\,\mathrm{sum}(\mathrm{sum}((D*(O\odot O\odot O\odot O)*(D\odot D))\odot R)) \tag{100}$$

$$\gamma(E_8) = 7\,\mathrm{sum}(\mathrm{sum}(((O\odot O)*(O\odot O)*(D\odot D\odot D))\odot R)) \tag{101}$$

$$\gamma(E_9) = 14\,\mathrm{sum}(\mathrm{sum}(((O\odot O)*(D\odot D\odot D)*(O\odot O))\odot R)) \tag{102}$$

$$\gamma(E_{10}) = 7\,\mathrm{sum}(\mathrm{sum}(((O\odot O\odot O\odot O)*(O\odot O)*D)\odot R)) \tag{103}$$

$$\gamma(E_{11}) = 21\,\mathrm{sum}(\mathrm{sum}(((O\odot O\odot O\odot O)*D*(O\odot O))\odot R)) \tag{104}$$

$$\gamma(E_{12}) = 14\,\mathrm{sum}(\mathrm{sum}((D*(O\odot O\odot O\odot O)*(O\odot O))\odot R)) \tag{105}$$

$$\gamma(E_{13}) = 7\,\mathrm{tr}((O\odot O\odot O\odot O\odot O)*O*O) \tag{106}$$

$$\gamma(E_{14}) = 14\,\mathrm{tr}((O\odot O\odot O)*O*(O\odot O\odot O)) \tag{107}$$

$$\gamma(E_{15}) = 7\,\mathrm{sum}(\mathrm{sum}(((O\odot O)*(O\odot O))\odot R)\odot\mathrm{sum}((O\odot O)*D) - \mathrm{sum}(((O\odot O\odot O\odot O)*D*(O\odot O))\odot R)$$
$$-\mathrm{diag}(((O\odot O)*D*(O\odot O)*(O\odot O)))) \tag{108}$$

$$\gamma(E_{16}) = 14\,\mathrm{sum}((\mathrm{sum}(((O\odot O)*(O\odot O))\odot R)\odot\mathrm{sum}((O\odot O)) - \mathrm{sum}(((O\odot O\odot O\odot O)*(O\odot O))\odot R)$$
$$-\mathrm{diag}(((O\odot O)*(O\odot O)*(O\odot O))))\odot\mathrm{diag}(D)) \tag{109}$$

$$\gamma(E_{17}) = 7\,\mathrm{sum}(((\mathrm{sum}(O\odot O)\odot\mathrm{sum}(O\odot O)\odot\mathrm{sum}(O\odot O)) - \mathrm{sum}((O\odot O\odot O\odot O\odot O\odot O))$$
$$-3\,(\mathrm{sum}((O\odot O\odot O\odot O))\odot\mathrm{sum}((O\odot O)) - \mathrm{sum}((O\odot O\odot O\odot O\odot O\odot O)))))\odot\mathrm{diag}(D)) \tag{110}$$

$$Z_1 \equiv 0.5\,((\mathrm{sum}(O\odot O)\odot\mathrm{sum}(O\odot O)) - \mathrm{sum}((O\odot O\odot O\odot O)))$$

$$\gamma(E_{18}) = 14\,\mathrm{sum}(\mathrm{sum}((O\odot O)*D)\odot Z_1 - \mathrm{sum}((O\odot O\odot O\odot O)*D)\odot\mathrm{sum}((O\odot O))$$
$$+\mathrm{sum}((O\odot O\odot O\odot O\odot O\odot O)*D)) \tag{111}$$

$$\gamma(E_{19}) = 28\,\mathrm{sum}(\mathrm{diag}((O\odot O)*(O\odot O)*(O\odot O))\odot\mathrm{diag}(D)) \tag{112}$$

$$\gamma(E_{20}) = 21\,\mathrm{sum}(\mathrm{sum}((D*(O\odot O)*(D\odot D)*(O\odot O))\odot R)) \tag{113}$$

$$\gamma(E_{21}) = 14\,\mathrm{sum}(\mathrm{sum}(((D\odot D)*(O\odot O)*D*(O\odot O))\odot R)) \tag{114}$$

$$\gamma(E_{22}) = 7\,\mathrm{sum}(\mathrm{sum}((D*(O\odot O)*(O\odot O)*(D\odot D))\odot R)) \tag{115}$$

$$\gamma(E_{23}) = 7\,\mathrm{sum}(\mathrm{diag}(O*O*O)\odot\mathrm{diag}((D\odot D\odot D\odot D))) \tag{116}$$

$$\gamma(E_{24}) = 28\,\mathrm{sum}(\mathrm{diag}((O\odot O\odot O)*O*O)\odot\mathrm{sum}((O\odot O)) - \mathrm{diag}((O\odot O\odot O\odot O\odot O)*O*O)$$
$$-\mathrm{diag}((O\odot O\odot O)*O*(O\odot O\odot O))) \tag{117}$$

$$\gamma(E_{25}) = 7\,\mathrm{sum}(\mathrm{diag}(O*(O\odot O\odot O)*O)\odot\mathrm{sum}((O\odot O)) - 2\,\mathrm{diag}((O\odot O\odot O)*(O\odot O\odot O)*O)) \tag{118}$$

$$\gamma(E_{26}) = 7\,\mathrm{sum}(\mathrm{diag}(O*(O\odot O\odot O)*O)\odot\mathrm{diag}((D\odot D))) \tag{119}$$

$$\gamma(E_{27}) = 42\,\mathrm{sum}(\mathrm{diag}((O\odot O\odot O)*O*O)\odot\mathrm{diag}((D\odot D))) \tag{120}$$

$$\gamma(E_{28}) = 7\,\mathrm{sum}(\mathrm{diag}(O*O*O)\odot\mathrm{sum}((O\odot O\odot O\odot O\odot O)) - 2\,\mathrm{diag}((O\odot O\odot O\odot O\odot O)*O*O)) \tag{121}$$

$$\gamma(E_{29}) = 7\,\mathrm{sum}(\mathrm{sum}((D*(O\odot O)*D*(O\odot O)*D)\odot R)) \tag{122}$$

$$\gamma(E_{30}) = 28\,\mathrm{sum}(\mathrm{diag}(O*D*(O\odot O\odot O)*O)\odot\mathrm{diag}(D)) \tag{123}$$

$$\gamma(E_{31}) \quad = \quad 28\,\mathrm{tr}(O*D*(O\odot O\odot O)*D*O) \tag{124}$$

$$\gamma(E_{32}) \quad = \quad 14\,\mathrm{sum}(\mathrm{diag}(O*(D\odot D)*O*O)\odot\mathrm{sum}((O\odot O)) - \mathrm{diag}((O\odot O\odot O)*O*(D\odot D)*O)$$
$$-\mathrm{diag}((O\odot O\odot O)*(D\odot D)*O*O)) \tag{125}$$

$$\gamma(E_{33}) \quad = \quad 14\,\mathrm{sum}(\mathrm{diag}(O*D*O*O)\odot\mathrm{diag}((D\odot D\odot D))) \tag{126}$$

$$\gamma(E_{34}) \quad = \quad 7\mathrm{tr}(O*(D\odot D)*O*(D\odot D)*O) \tag{127}$$

$$\gamma(E_{35}) \quad = \quad 7(\mathrm{sum}(\mathrm{sum}((((O*O)\odot R)\odot((O*(D\odot D\odot D)*O)\odot R))))$$
$$-\mathrm{sum}(\mathrm{sum}(((O\odot O)*(D\odot D\odot D)*(O\odot O))\odot R))) \tag{128}$$

$$\gamma(E_{36}) \quad = \quad 14\,\mathrm{sum}(\mathrm{sum}(((O\odot O\odot O)*O)\odot R\odot(O*D*O))$$
$$-\mathrm{sum}(((O\odot O\odot O\odot O)*D*(O\odot O))\odot R)) \tag{129}$$

$$\gamma(E_{37}) \quad = \quad 28\,\mathrm{sum}(\mathrm{sum}(((O\odot O\odot O)*D*O)\odot R\odot(O*O))$$
$$-\mathrm{sum}(((O\odot O\odot O\odot O\odot O)*D*(O\odot O))\odot R)) \tag{130}$$

$$Z_2 \quad \equiv \quad (((O*O)\odot R)*O - O\odot(\mathbb{1}_{d\times 1}*(\mathrm{sum}((O\odot O)^{\top}))^{\top} - (O\odot O)))\odot R \tag{131}$$

$$Z_3 \quad \equiv \quad (O\odot((O*O)\odot R))\odot R \tag{132}$$

$$Z_4 \quad \equiv \quad (O\odot(((O\odot O\odot O\odot O\odot O)*O)\odot R))\odot R \tag{133}$$

$$Z_6 \quad \equiv \quad ((O\odot O\odot O)\odot((O*O)\odot R))\odot R \tag{134}$$

$$Z_7 \quad \equiv \quad (O\odot(((O\odot O\odot O)*(O\odot O\odot O))\odot R))\odot R \tag{135}$$

$$\gamma(E_{38}) \quad = \quad 7\,\mathrm{sum}(\mathrm{sum}((((O\odot O\odot O)*O)\odot R\odot Z_2 - (((O\odot O\odot O\odot O)*Z_3)\odot R - Z_4)$$
$$-((Z_6*(O\odot O))\odot R - Z_7)))) \tag{136}$$

$$Z_7 \quad \equiv \quad 0.5\,\mathrm{sum}(\mathrm{sum}(O\odot(((O\odot O)*(O\odot O))\odot R)\odot((O*O)\odot R)$$
$$-O\odot(((O\odot O\odot O)*(O\odot O\odot O))\odot R))) \tag{137}$$

$$\gamma(E_{39}) \quad = \quad 7\,(\mathrm{sum}(\mathrm{sum}((O\odot((O*O)\odot R)\odot(\mathrm{sum}((O\odot O))*\mathbb{1}_{1\times d}$$
$$-(O\odot O))\odot(\mathbb{1}_{d\times 1}*(\mathrm{sum}((O\odot O)^{\top}))^{\top} - (O\odot O)))))$$
$$-\mathrm{sum}(\mathrm{sum}((O\odot(((O\odot O\odot O)*O)\odot R)\odot(\mathbb{1}_{d\times 1}*(\mathrm{sum}((O\odot O)^{\top}))^{\top} - (O\odot O)))))$$
$$-\mathrm{sum}(\mathrm{sum}((O\odot((O*(O\odot O\odot O))\odot R)\odot(\mathrm{sum}((O\odot O))*\mathbb{1}_{1\times d} - (O\odot O)))))$$
$$+\mathrm{sum}(\mathrm{sum}((O\odot(((O\odot O\odot O)*(O\odot O\odot O))\odot R)))))) - 14\,Z_7 \tag{138}$$

$$\gamma(E_{40}) \quad = \quad 21\,\mathrm{sum}(\mathrm{diag}((D\odot D)*O*O*O)\odot\mathrm{sum}((O\odot O)) - 2\,\mathrm{diag}((D\odot D)*(O\odot O\odot O)*O*O)) \tag{139}$$

$$\gamma(E_{41}) \quad = \quad 7\,\mathrm{sum}(\mathrm{diag}(O*O*O)\odot\mathrm{sum}((O\odot O)*(D\odot D)) - 2\,\mathrm{diag}((O\odot O\odot O)*(D\odot D)*O*O)) \tag{140}$$

$$\gamma(E_{42}) \quad = \quad 7\,(\mathrm{sum}(\mathrm{diag}(O*O*O)\odot\mathrm{sum}(((O\odot O)*(O\odot O))\odot R) - 2\,\mathrm{diag}((O\odot O\odot O)*(O\odot O\odot O)*O))$$
$$-2\,\mathrm{sum}(\mathrm{diag}((O\odot O\odot O)*O*O)\odot\mathrm{sum}((O\odot O)) - \mathrm{diag}((O\odot O\odot O\odot O\odot O)*O*O)$$
$$-\mathrm{diag}((O\odot O\odot O)*O*(O\odot O\odot O)))) - 28\,Z_7 \tag{141}$$

$$\gamma(E_{43}) \quad = \quad 14\,\mathrm{sum}(\mathrm{diag}(O*O*O)\odot Z_1 - 2\,(\mathrm{diag}((O\odot O\odot O)*O*O)\odot\mathrm{sum}((O\odot O))$$
$$-\mathrm{diag}((O\odot O\odot O\odot O\odot O)*O*O) - 0.5\,\mathrm{diag}((O\odot O\odot O)*O*(O\odot O\odot O)))) \tag{142}$$

$$\gamma(E_{44}) \quad = \quad 56\,Z_7 \tag{143}$$

$$Z_8 \quad \equiv \quad (O\odot(((O\odot O\odot O)*O)\odot R))\odot R \tag{144}$$

$$Z_9 \quad \equiv \quad (O\odot((O*O)\odot R))\odot R \tag{145}$$

$$Z_{10} \quad \equiv \quad (O\odot((O*(O\odot O\odot O))\odot R))\odot R \tag{146}$$

$$Z_{11} \quad \equiv \quad ((O*O)\odot R\odot Z_2 - (((O\odot O)*Z_3)\odot R - Z_8) - ((Z_9*(O\odot O))\odot R - Z_{10})) \tag{147}$$

$$\gamma(E_{45}) \quad = \quad 14\,(\mathrm{sum}(0.5\,\mathrm{sum}(Z_{11})\odot\mathrm{sum}((O\odot O))) - (1/7)\,\gamma(E_{38}) - \mathrm{sum}(\mathrm{sum}(((O\odot O)\odot Z_{11}))) \tag{148}$$

$$\gamma(E_{46}) = 21 \operatorname{sum}(\operatorname{sum}(((O\odot O))\odot Z_{11})) \tag{149}$$

$$\gamma(E_{47}) = 7 \operatorname{sum}(\operatorname{sum}(Z_{11}\odot\operatorname{diag}((D\odot D)))) \tag{150}$$

$$\gamma(E_{48}) = 7 \operatorname{tr}((D\odot D)*O*D*O*D*O) \tag{151}$$

$$\gamma(E_{49}) = 14 \operatorname{sum}(\operatorname{diag}(D*O*O*O)\odot\operatorname{sum}((O\odot O)*D) - 2 \operatorname{diag}(D*(O\odot O\odot O)*D*O*O)) \tag{152}$$

$$\gamma(E_{50}) = 14 \operatorname{sum}(\operatorname{diag}(O*O*D*O)\odot\operatorname{sum}((O\odot O)*D) - \operatorname{diag}((O\odot O\odot O)*D*O*D*O)$$
$$-\operatorname{diag}((O\odot O\odot O)*(D\odot D)*O*O)) \tag{153}$$

$$\gamma(E_{51}) = 28 \operatorname{sum}(\operatorname{diag}(D*O*D*O*O)\odot\operatorname{sum}((O\odot O)) - \operatorname{diag}(D*(O\odot O\odot O)*D*O*O)$$
$$-\operatorname{diag}(D*(O\odot O\odot O)*O*D*O)) \tag{154}$$

$$\gamma(E_{52}) = 7 \operatorname{sum}(\operatorname{diag}(O*D*O*D*O)\odot\operatorname{sum}((O\odot O)) - 2 \operatorname{diag}((O\odot O\odot O)*D*O*D*O)) \tag{155}$$

$$\gamma(E_{53}) = 14 \operatorname{sum}((\operatorname{sum}(((((O*O)\odot R)\odot((O*D*O)\odot R))$$
$$-((O\odot O)*D*(O\odot O))\odot R)))\odot\operatorname{diag}((D\odot D))) \tag{156}$$

$$\gamma(E_{54}) = 7 \operatorname{sum}(\operatorname{sum}(((((O*D*O)\odot R)\odot((O*(D\odot D)*O)\odot R))$$
$$-((O\odot O)*(D\odot D\odot D)*(O\odot O))\odot R))) \tag{157}$$

$$Z_{12} \equiv \operatorname{sum}(0.5 \operatorname{sum}(((((O*O)\odot R)\odot((O*O)\odot R))$$
$$-(((O\odot O)*(O\odot O))\odot R))\odot((O\odot O)*D))) \tag{158}$$

$$Z_{13} \equiv \operatorname{sum}(\operatorname{sum}((((((O\odot O\odot O)*D*O)\odot R)\odot((O*O)\odot R))$$
$$-((O\odot O\odot O\odot O)*D*(O\odot O))\odot R))) \tag{159}$$

$$Z_{14} \equiv 0.5 \operatorname{sum}(\operatorname{sum}(((((O*D*O)\odot R)\odot((O*O)\odot R))$$
$$-(((O\odot O)*D*(O\odot O))\odot R))\odot((O\odot O)))) \tag{160}$$

$$Z_{15} \equiv \operatorname{sum}(\operatorname{sum}((((((O\odot O\odot O)*O)\odot R)\odot((O*D*O)\odot R))$$
$$-((O\odot O\odot O\odot O)*D*(O\odot O))\odot R))) \tag{161}$$

$$\gamma(E_{55}) = 14 (\operatorname{sum}(0.5 (\operatorname{sum}(((((O*O)\odot R)\odot(((O*O))\odot R))$$
$$-((O\odot O)*(O\odot O))\odot R)))\odot\operatorname{sum}((O\odot O)*D)) - Z_{13} - Z_{12}) \tag{162}$$

$$\gamma(E_{56}) = 28 (\operatorname{sum}(0.5 (\operatorname{sum}(((((O*O)\odot R)\odot(((O*O))\odot R))$$
$$-((O\odot O)*(O\odot O))\odot R)))\odot\operatorname{sum}(D*(O\odot O))) - Z_{13} - Z_{12}) \tag{163}$$

$$\gamma(E_{57}) = 14 (\operatorname{sum}((\operatorname{sum}(((((O*D*O)\odot R)\odot(((O*O))\odot R))$$
$$-((O\odot O)*D*(O\odot O))\odot R)))\odot\operatorname{sum}((O\odot O))) - Z_{13} - Z_{15} - 2 Z_{14}) \tag{164}$$

$$\gamma(E_{58}) = 14 (\operatorname{sum}(0.5 \operatorname{sum}((((((O*O)\odot R)\odot((O*O)\odot R))$$
$$-(((O\odot O)*(O\odot O))\odot R))*D))\odot\operatorname{sum}((O\odot O))) - Z_{15} - Z_{12}) \tag{165}$$

$$\gamma(E_{59}) = 84 Z_{12} \tag{166}$$

$$\gamma(E_{60}) = 42 Z_{14} \tag{167}$$

$$Z_{25} = \operatorname{tr}(M*M*M*M*M*M*M) - \operatorname{sum}(\gamma(E_1) : \gamma(E_{60})) \tag{168}$$

$$Z_{26} = \operatorname{tr}(O*O*O*O*O*O*O) - \gamma(E_{13}) - \gamma(E_{14}) - \gamma(E_{24})$$
$$-\gamma(E_{25}) - \gamma(E_{28}) - \gamma(E_{38}) - \gamma(E_{39}) - \operatorname{sum}(\gamma(E_{42}) : \gamma(E_{46})) \tag{169}$$

$$Z_{16} \equiv (1/6) ((O*O\odot R)\odot(O*O\odot R)\odot(O*O\odot R) - ((O\odot O\odot O)*(O\odot O\odot O)\odot R)$$
$$-3 (((O\odot O)*(O\odot O)\odot R)\odot(O*O\odot R) - ((O\odot O\odot O)*(O\odot O\odot O)\odot R))) \tag{170}$$

$$\gamma(E_{61}) = 42 \operatorname{sum}(\operatorname{sum}(Z_{16}\odot O)) \tag{171}$$

$$Z_{17} \equiv \operatorname{sum}(\operatorname{sum}(0.5\,((O{*}O\odot R)\odot(O{*}O\odot R)$$
$$-((O\odot O){*}(O\odot O)\odot R)))\odot(0.5\,\operatorname{diag}(O{*}O{*}O))) \tag{172}$$

$$\gamma(E_{62}) = 28\,(Z_{17} - (6/84)\,\gamma(E_{61}) - (2/42)\,\gamma(E_{46}) - (3/56)\,\gamma(E_{44}) \tag{173}$$

$$\gamma(E_{63}) = Z_{26} - \gamma(E_{61}) - \gamma(E_{62}) \tag{174}$$

$$\gamma(E_{64}) = 7\operatorname{sum}(\operatorname{sum}((D{*}O{*}D{*}O{*}D\odot R)\odot(O{*}O\odot R)))$$
$$-7\operatorname{sum}(\operatorname{sum}(D{*}(O\odot O){*}D{*}(O\odot O){*}D\odot R)) \tag{175}$$

$$\gamma(E_{65}) = 7\operatorname{sum}(\operatorname{sum}(D{*}Z_{11}{*}D)) \tag{176}$$

$$Z_{18} \equiv \operatorname{sum}(((O{*}O)\odot R\odot(O{*}O) - ((O\odot O){*}(O\odot O))\odot R)\odot((O\odot O))) \tag{177}$$

$$\gamma(E_{66}) = 7\operatorname{sum}(((\operatorname{diag}(O{*}O{*}O)\odot\operatorname{diag}(O{*}O{*}O))$$
$$-2\,\operatorname{diag}((O\odot O){*}(O\odot O){*}(O\odot O)) - 4\,Z_{18})\odot\operatorname{diag}(D)) \tag{178}$$

$$Z_{20} \equiv 0.5\operatorname{sum}(\operatorname{sum}(((O{*}O\odot R)\odot(O{*}D{*}O\odot R) - ((O\odot O){*}D{*}(O\odot O)))\odot(O\odot O))) \tag{179}$$

$$\gamma(E_{67}) = 14\,(\operatorname{sum}(\operatorname{diag}(O{*}O{*}O)\odot\operatorname{diag}(O{*}O{*}D{*}O) - 2\,\operatorname{diag}((O\odot O){*}(O\odot O){*}D{*}(O\odot O)))$$
$$-2\operatorname{sum}(Z_{18}\odot\operatorname{diag}(D)) - 4\,Z_{20}) \tag{180}$$

$$Z_{21} \equiv (((O{*}D{*}O{*}D)\odot R){*}O - O\odot(\mathbb{1}_{d\times 1}{*}\operatorname{sum}(D{*}(O\odot O){*}D,1)$$
$$-D{*}(O\odot O){*}D))\odot R \tag{181}$$

$$Z_{22} \equiv (O\odot((O{*}D{*}O)\odot R))\odot R \tag{182}$$

$$Z_{23} \equiv (O\odot((D{*}(O\odot O\odot O){*}D{*}O)\odot R))\odot R \tag{183}$$

$$Z_{24} \equiv (O\odot((O{*}D{*}(O\odot O\odot O){*}D)\odot R))\odot R \tag{184}$$

$$\gamma(E_{68}) = 7\operatorname{sum}(\operatorname{sum}(((O{*}O)\odot R\odot Z_{21} - (((O\odot O){*}D{*}Z_{22})\odot R - Z_{23})$$
$$-((Z_{22}{*}D{*}(O\odot O))\odot R - Z_{24})))) \tag{185}$$

$$\gamma(E_{69}) = Z_{25} - Z_{26} - \operatorname{sum}(\gamma(E_{64}):\gamma(E_{68})) \tag{186}$$