[Reviews · NeurIPS 2017]

Reviewer 1



The paper introduces a method for estimating Schatten k-norms of a matrix from very sparse samples of matrix entries. The paper focuses on the sampling density regime, where existing matrix completion techniques fail. The basic idea of this paper is to estimate the trace of M^k from samples of M. The trace of M^k is given by the sum of weights of k-cycles, where the weight of each k-cycle is given by the product of edge weights. The estimator for a sampled matrix is then modified to take the probability of a particular cycle being sampled. This probability, which depends on the type of the cycle, is given explicitly for all types of cycles when k <= 7. The paper provides stochastic bounds on the accuracy of the estimate. The method is effective on synthetic datasets. The overall quality of the paper is good (particularly the proof of Theorem 1). Below are a few comments regarding the current submission: 1) When d is significantly bigger than k, is it possible to directly obtain M^k and compute its trace? Matrix multiplication for sparse matrices is highly efficiently. Would Trace(M_0^k) provide a good estimate for Trace(M^{k}) when M_0 is sampled from M? If so, would this be more efficient than $O(d^{2.373}$? 2) Alternatively, when k is even, trace(M^k) is the same as the squared sum of the entries of M^{k/2}. Is it possible to estimate that from M_0^{k/2}? 3) The bound in theorem 1 seems to be not very tight. rho^2 is bigger than $(\kappa\mu)^{2\kappa}g(k)$, where $\kappa$ is the condition number, and \mu is the incoherence number. It seems to be \rho is fairly bigger. If \rho is bigger then, r^{1-k/2} has to be significantly smaller than $d\cdot p$, in order for the current bound to be useful. I would like to see comments on this. It is good to given a simplified bound on some simple matrices, e.g., the matrix whose entries are 1, or something similar. 4) $d$ is usually reserved for vertex degree, and it is recommended to use $n$ and $m$ for matrix dimensions. 5) Are there any real examples? I would like to see how to address these issues in the rebuttal.

Reviewer 2



Summary of paper: The paper proposes an estimator for the Schatten norm of a matrix when only few entries of the matrix is observed. The estimator is based on the relation between the Schatten k-norm of a matrix and the total weight of all closed k-length walks on the corresponding weighted graph. While the paper restricts the discussion to symmetric positive semi-definite matrices, and guarantees given for uniform sampling, the generic principle is applicable for any matrix and any sampling strategy. The results hold under standard assumptions, and corollaries related to spectrum and rank estimation are also provided in the appendix. Clarity: The paper is very well written and organised, including some of the material in the supplementary. The motivation for the work, and the discussion that builds up to the estimator and well presented. The numerical simulations are also perfectly positioned. Quality: The technical quality of the paper is high. The main structure of the proofs are correct, but I didn’t verify the computations in equations (26)-(36). The most pleasing component of the theoretical analysis is the level at which it is presented. The authors restrict the discussion to symmetric psd matrices for ease of presentation, but the sampling distribution is kept general until the final guarantees in Section 3.1. The authors also correctly note estimating the Schatten norm itself is never the end goal, and hence, they provide corollaries related to spectrum and rank estimation in the appendix. The numerical simulations do not involve a real application, but clearly show the advantage of the proposed estimator over other techniques when sampling rate is very small. That being said, I feel that further numerical studies based on this approach is important. Significance: The paper considers a classical problem that lies at the heart of several matrix related problems. Hence, the paper is very relevant for the machine learning community. The generality at which the problem is solved, or its possible extensions, has several potential applications. To this end, I feel that it will be great if the authors can follow up this work with a longer paper that also includes the more complicated case of rectangular matrices, analyses other sampling strategies (like column sampling / sub matrix sampling), and more thorough experimental studies. Originality: Since the underlying problem is classical, one should consider that many ideas have been tried out in the literature. I feel that the individual components of the discussion and analysis are fairly standard techniques (except perhaps the computational simplification for k<=7). But I feel that the neat way in which the paper combines everything is very original. For instance, the use of polynomial concentration in the present context is intuitive, but the careful computation of the variation bound is definitely novel. Minor comments: 1. Line 28-29: Can we hope to estimate the spectrum of a matrix from a submatrix? I am not very optimistic about this. 2. Appendix B: It would be nice if a description of this method is added. In particular, what is the time complexity of the method? 3. Line 238,262,281,292: In standard complexity terminology, I think d^2p = O(something) should actually be d^2p = \Omega(something). Big-O usually means at most upto constant, whereas here I think you would like to say that the number of observed entries is at least something (Big-Omega). 4. Line 263: How do you get the condition on r? The third term in (8) is less than 1 if r is less than d^(1-1/k).

Reviewer 3



This paper presents a new estimator which is effective to predict the matrix schatten norm. The main technique is constructing a k-cyclic pseudograph to fit the statistics of the matrix. Though the quantity "k" is restricted to be smaller than 8, it seems not a big hurt to the contribution since many applications involve only "k = 1 or k = 2". I would like to see more discussion on how to extend the results to general non-symmetric matrices. Authors claimed that the algorithm can naturally be generalized, but I wonder whether the theoretical results still hold. It is also worth mentioning the major difference from [2] and [12].